# The MODY-associated *KCNK16* L114P mutation increases islet glucagon secretion and limits insulin secretion resulting in transient neonatal diabetes and glucose dyshomeostasis in adults

Arya Y Nakhe[1], Prasanna K Dadi[1], Jinsun Kim[1,2], Matthew T Dickerson[1], Soma Behera[1], Jordyn R Dobson[1], Shristi Shrestha[3], Jean-Philippe Cartailler[3], Leesa Sampson[3], Mark A Magnuson[1,3,4], David A Jacobson[1]*

[1]Department of Molecular Physiology and Biophysics, Vanderbilt University, Nashville, United States; [2]Department of Chemistry, Vanderbilt University, Nashville, United States; [3]Center for Stem Cell Biology, Vanderbilt University, Nashville, United States; [4]Department of Cell and Developmental Biology, Vanderbilt University, Nashville, United States

*For correspondence: david.a.jacobson@vanderbilt.edu

Competing interest: The authors declare that no competing interests exist.

**Abstract** The gain-of-function mutation in the TALK-1 K⁺ channel (p.L114P) is associated with maturity-onset diabetes of the young (MODY). TALK-1 is a key regulator of β-cell electrical activity and glucose-stimulated insulin secretion. The *KCNK16* gene encoding TALK-1 is the most abundant and β-cell-restricted K⁺ channel transcript. To investigate the impact of *KCNK16* L114P on glucose homeostasis and confirm its association with MODY, a mouse model containing the *Kcnk16* L114P mutation was generated. Heterozygous and homozygous *Kcnk16* L114P mice exhibit increased neonatal lethality in the C57BL/6J and the CD-1 (ICR) genetic background, respectively. Lethality is likely a result of severe hyperglycemia observed in the homozygous *Kcnk16* L114P neonates due to lack of glucose-stimulated insulin secretion and can be reduced with insulin treatment. *Kcnk16* L114P increased whole-cell β-cell K⁺ currents resulting in blunted glucose-stimulated Ca²⁺ entry and loss of glucose-induced Ca²⁺ oscillations. Thus, adult *Kcnk16* L114P mice have reduced glucose-stimulated insulin secretion and plasma insulin levels, which significantly impairs glucose homeostasis. Taken together, this study shows that the MODY-associated *Kcnk16* L114P mutation disrupts glucose homeostasis in adult mice resembling a MODY phenotype and causes neonatal lethality by inhibiting islet insulin secretion during development. These data suggest that TALK-1 is an islet-restricted target for the treatment for diabetes.

## eLife assessment

This study characterizes how a point mutation in the TALK-1 potassium channel, encoded by the KCNK16 gene, causes MODY diabetes. The mutation, L114P, causes a gain-of-function to increase K+ currents and inhibit glucose-stimulated insulin secretion. Increased glucagon likely results from paracrine effects in the islets. The data are **convincing** and the work will be **valuable** for understanding islet function.

## Introduction

Maturity-onset diabetes of the young (MODY) is a collection of monogenic forms of early-onset familial diabetes resulting from β-cell dysfunction. To date, mutations in 15 genes involved in β-cell development and function have been associated with MODY (*Broome et al., 2021*). Monogenic diabetes cases account for approximately 1–5% of the total diabetic patient population (*Kavvoura and Owen, 2012*; *Zhang et al., 2021*). However, as the phenotype of MODY overlaps with other forms of diabetes, many patients are misdiagnosed (*Shields et al., 2010*; *Greeley et al., 2022*). Data from monogenic diabetes registries also suggests that in the US, ~70% of the registered cases still do not have a known genetic cause, and similarly many patients in the UK (~50%) with monogenic diabetes have MODY-causing mutations that remain to be determined (*Bowden et al., 2021*; *Shepherd et al., 2016*). Additionally, some MODY-associated mutations are only reported in single families, and thus require genetic and mechanistic confirmation (*Yamagata et al., 1996*; *Pearson et al., 2005*). We recently identified a mutation in *KCNK16*, the gene encoding TALK-1 channels (p. *Kcnk16* L114P), which co-segregates with MODY in a four-generation family (*Graff et al., 2021*). As *KCNK16* is the most islet-restricted and abundant β-cell K$^+$ channel transcript, the *Kcnk16* L114P mutation is likely to perturb β-cell function and cause MODY (*Bramswig et al., 2013*; *Blodgett et al., 2015*). This was further strengthened by a recent report of a Japanese family with the identical *KCNK16* mutation (p. *Kcnk16* L114P) segregating with MODY (*Katsuyuki Matsui et al., 2023*). Thus, it is important to determine the mechanistic underpinnings of how this specific mutation in TALK-1 impacts islet function and results in glucose dyshomeostasis.

TALK-1 is a key regulator of β-cell membrane potential ($V_m$), glucose-stimulated Ca$^{2+}$ influx, and insulin secretion (*Vierra et al., 2015*). A non-synonymous gain-of-function polymorphism in *KCNK16* (rs1535500; p. *Kcnk16* A277E) is also associated with an increased risk for type 2 diabetes (T2D) (*Cho et al., 2011*; *Mahajan et al., 2014*). Moreover, we recently determined that the MODY-associated *Kcnk16* L114P mutation results in a significant gain-of-function in TALK-1 K$^+$ flux (*Graff et al., 2021*). Thus, there is strong genetic evidence that alterations in TALK-1 function (e.g. *Kcnk16* L114P and *Kcnk16* A277E) result in diabetic phenotypes. Interestingly, when heterologously expressed in β-cells, TALK-1 L114P inhibited glucose-stimulated membrane potential ($V_m$) depolarization and Ca$^{2+}$ influx in most β-cells. Although TALK-1 L114P-mediated changes in inhibition of electrical activity and Ca$^{2+}$ influx would be predicted to fully inhibit glucose-stimulated insulin secretion, they only resulted in partial blunting of β-cell insulin secretion. However, this is consistent with *Kcnk16* L114P MODY patients that required low-dose insulin therapy which also suggested that this mutation does not fully suppress glucose-stimulated insulin secretion. This differs from previous studies showing that inhibition of β-cell Ca$^{2+}$ influx with either K$^+$ channel pharmacological activation or gain-of-function mutations (e.g. K$_{ATP}$ R201H) results in complete inhibition of glucose-stimulated insulin secretion (*Gloyn et al., 2004*). Therefore, it is critical to determine if endogenous TALK-1 L114P expression also shows complete inhibition of β-cell Ca$^{2+}$ influx and how this impacts glucose-stimulated insulin secretion. Another potential mechanism for the modest impact of *Kcnk16* L114P on insulin secretion could be due to its function in other islet cells. For example, TALK-1 is also expressed in δ-cells where its activity limits somatostatin secretion (*Vierra et al., 2018*). Because somatostatin exerts inhibitory tone on islet β- and α-cells, TALK-1 L114P-mediated reductions in δ-cell somatostatin secretion would be predicted to increase both glucagon and insulin secretion. Due to the β-cell-intrinsic role of TALK-1 L114P channels, the effect of somatostatin on glucose-stimulated insulin secretion might be limited; however, it remains to be determined if the fasting hyperglycemia observed in *KCNK16*-MODY patients is due in part to elevated glucagon secretion.

The nature and severity of MODY phenotypes is dictated by how specific gene mutations affect β-cell function. K$_{ATP}$ is the only ion channel besides TALK-1 to be linked to MODY; this is due to mutations in genes encoding the K$_{ATP}$ channel complex (*KCNJ11* and *ABCC8*) or mutations in genes affecting ATP synthesis (e.g. glucokinase) (*Bonnefond et al., 2012*; *Huopio et al., 2003*). Interestingly, MODY-associated mutations in *KCNJ11* and *ABCC8* have also been found to cause other diabetic phenotypes including permanent or transient neonatal diabetes, and late-onset diabetes (*Yorifuji et al., 2005*; *Devaraja et al., 2020*). Although the two families with *Kcnk16* L114P exhibit a MODY phenotype, it remains to be determined if *KCNK16* mutations are associated with other diabetic phenotypes besides MODY. Indeed, it was originally predicted that mutations that cause a substantial gain-of-function in TALK-1 channel activity (e.g. *Kcnk16* L114P) would act similarly to

the gain-of-function mutations in $K_{ATP}$ subunits that cause neonatal diabetes (*Vierra et al., 2015*). However, the biophysical activity of TALK-1 channels differ in many ways from $K_{ATP}$ channels, and this could contribute to more modest MODY phenotype observed in patients with *KCNK16* L114P. For example, depolarization-dependent activation (outward rectification) of TALK-1 currents may limit the activity of TALK-1 L114P channels at resting $V_m$ (*Girard et al., 2001*). $K_{ATP}$ channels also show larger unitary conductance than TALK-1, not much voltage dependence, and are nucleotide gated (*Rorsman and Ashcroft, 2018*; *Kang and Kim, 2004*). The unique biophysical properties of $K_{ATP}$ and TALK-1 channels predict mechanistic differences in their corresponding gain-of-function phenotypes. However, it remains to be elucidated how endogenous *Kcnk16* L114P disrupts islet electrical activity and glucose homeostasis.

Here, we developed a mouse model harboring the MODY-associated *Kcnk16* variant p. *Kcnk16* L114P to investigate the impact of this mutation on glucose homeostasis and confirm its association with MODY. Interestingly, we observe increased neonatal lethality in heterozygous and homozygous *Kcnk16* L114P mice in the C57BL/6J (B6) and the B6;CD-1 (ICR) mixed genetic backgrounds, respectively, likely due to severe hyperglycemia and lack of glucose-stimulated insulin secretion. Whereas in adult mice, *Kcnk16* L114P blunts glucose-stimulated β-cell electrical activity, $Ca^{2+}$ handling, and glucose-stimulated insulin secretion, thus significantly impairing glucose tolerance. These data strongly suggest that alterations in TALK-1 activity can disrupt islet hormone secretion and glucose homeostasis. Importantly, this study further confirms that the *Kcnk16* L114P mutation results in a MODY phenotype, but additionally predicts that gain-of-function TALK-1 mutations cause transient neonatal diabetes.

## Results

### *Kcnk16* L114P neonates exhibit loss of glucose-stimulated $Ca^{2+}$ entry and insulin secretion leading to transient neonatal hyperglycemia and death

To confirm the association of *Kcnk16* L114P in causing MODY, a mouse model harboring the *Kcnk16* mutation was developed in the C57BL/6J background using CRISPR/spCas9 (B6 *Kcnk16* L114P; *Figure 1A* and *Figure 1—figure supplement 1A and B*). Surprisingly, heterozygous *Kcnk16* L114P (L/P) mice exhibited neonatal lethality as indicated by extremely low number of *Kcnk16* L114P (L/P) mice at weaning (*Figure 1B*). To increase the likelihood of survival of neonates, B6 *Kcnk16* L114P (L/P) mice were crossed with outbred CD-1 (ICR) mice resulting in a progeny in the mixed (50:50) background (B6;CD-1 *Kcnk16* L114P; *Figure 1A*). Intriguingly, in the mixed background, neonatal lethality was observed in the homozygous B6;CD-1 *Kcnk16* L114P (P/P) mice on ~postnatal day 4 (P4), but not in the heterozygous B6;CD-1 *Kcnk16* L114P (L/P) mice (*Figure 1—figure supplement 1C*). Lethality was likely independent of growth defects as body weight did not differ between genotypes (*Figure 1—figure supplement 1D*). To test if lethality occurs from gain-of-function TALK-1 L114P-mediated defect in neonatal islet function, we assessed glycemic control and insulin secretion on P4. Heterozygous and homozygous *Kcnk16* L114P neonates showed severe hyperglycemia and a concurrent reduction in plasma insulin levels in a gene dosage-dependent manner compared to the control littermates (WT) on P4 (*Figure 1C and D*, and *Figure 1—figure supplements 1E and 2A, B*). Hyperglycemia in neonatal *Kcnk16* L114P mice subsided by P10, when glycemia was equivalent to WT controls (*Figure 1—figure supplement 2*). Additionally, the *Kcnk16* L114P mutation blunted glucose-stimulated insulin secretion in P4 islets from heterozygous neonates and decreased it further in P4 islets from homozygous neonates (*Figure 1E and F*). These changes in glucose-stimulated insulin secretion and glucose homeostasis are likely due to TALK-1 L114P channel mediated alteration in islet function and not islet mass as pancreas weight was unchanged (*Figure 1G* and *Figure 1—figure supplement 1F*). Furthermore, neonatal islet composition was unaltered except for a modest increase in the α-cell (glucagon+) area/islet in the *Kcnk16* L114P (L/P) pancreas (*Figure 1H–J*). The loss of glucose-stimulated insulin secretion can be explained by a complete lack of glucose-stimulated $Ca^{2+}$ entry in islets from *Kcnk16* L114P (L/P and P/P) neonates (*Figure 1K–N*). However, these islets exhibited KCl-mediated β-cell $V_m$ depolarization-induced $Ca^{2+}$ entry; this indicates that TALK-1 L114P channels hyperpolarize β-cell $V_m$, thereby overriding the effect of $K_{ATP}$ closure due to glucose metabolism and lead to insufficient glucose-stimulated insulin secretion and hyperglycemia. Importantly, insulin

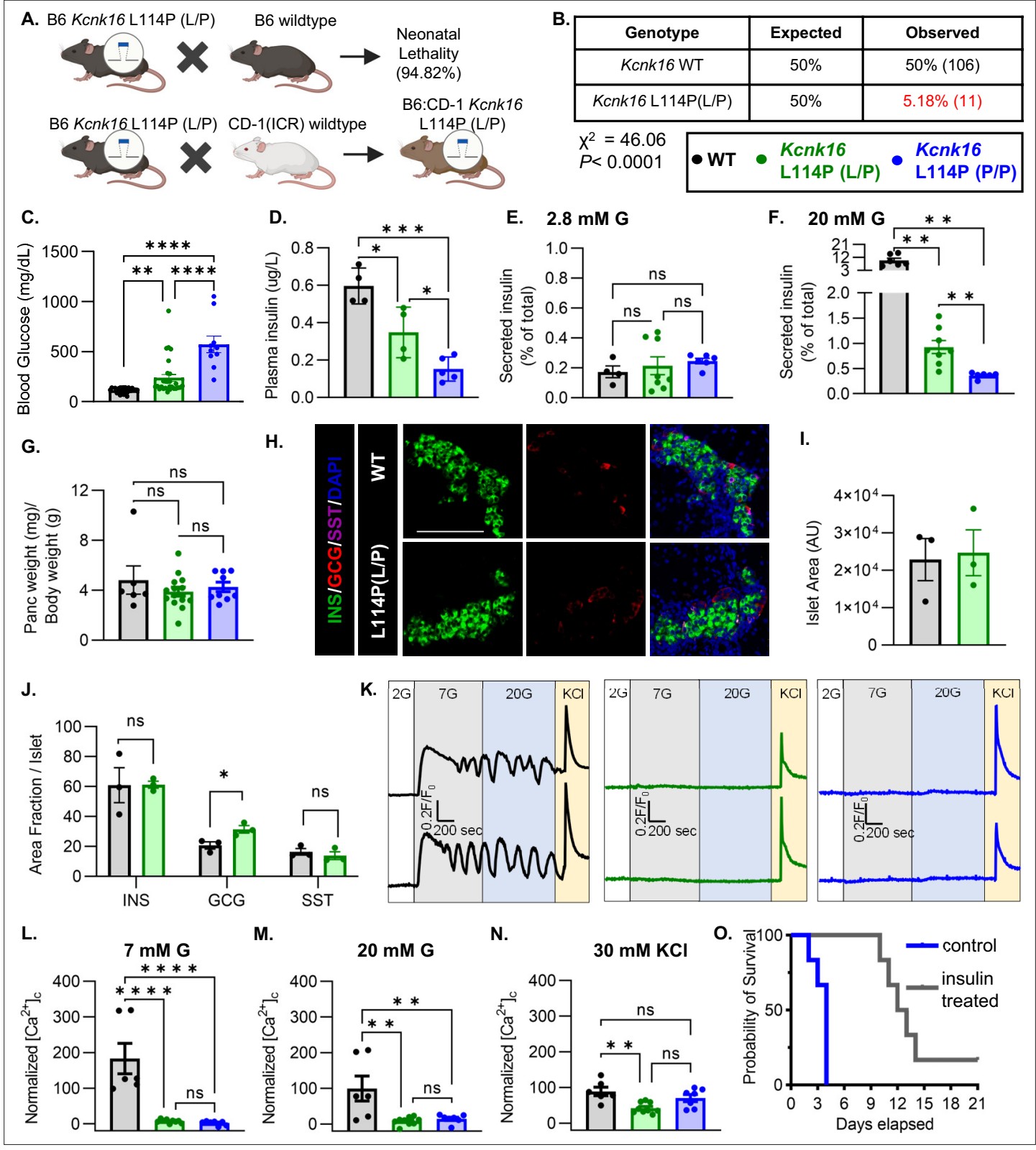

**Figure 1.** *Kcnk16*L114P neonates exhibit loss of glucose-stimulated Ca²⁺ entry and insulin secretion leading to transient neonatal hyperglycemia and death. (**A**) Schematic of the *Kcnk16* L114P mouse line generation in the C57BL/6J background and the C57BL/6J:CD-1 (ICR) mixed background. (**B**) $\chi^2$ analysis of the F1 progeny from C57BL/6J and heterozygous *Kcnk16* L114P (L/P) crossings genotyped at weaning on postnatal day 21 (P21). (**C**) Blood glucose measurements of male wildtype (WT; black; N=8), heterozygous *Kcnk16* L114P (L/P; green; N=16), and homozygous *Kcnk16* L114P (P/P; blue;

*Figure 1 continued on next page*

*Figure 1 continued*

N=9) mice on P4. (**D**) Plasma insulin measurements performed on P4 in WT (N=4), *Kcnk16* L114P (L/P; N=4), and *Kcnk16* L114P (P/P; N=5) neonates. (**E, F**) In vitro glucose-stimulated insulin secretion from P4 mouse islets stimulated with 2.8 mM glucose (**G**) or 20 mM G, respectively (WT; N=4, *Kcnk16* L114P (L/P); N=8, and *Kcnk16* L114P (P/P); N=6). (**G**) Pancreas weight/ body weight measurements of P4 male mice (WT; N=6, *Kcnk16* L114P (L/P); N=14, and *Kcnk16* L114P (P/P); N=9). (**H**) Representative immunostaining images of pancreas sections from P0 WT and *Kcnk16* L114P (L/P) mice (N=3 mice/ genotype), stained against insulin (INS; green), glucagon (GCG; red), somatostatin (SST; magenta), and Hoechst (blue); scale bar = 100 μm. (**I, J**) Average islet area and area fraction of hormone staining per islet quantified using Fiji ImageJ software in P0 mouse pancreas sections. (**K**) Representative glucose-stimulated Ca$^{2+}$ influx traces from P4 mouse islets sequentially stimulated with 2 mM glucose (**G**), 7 mM G, 20 mM G, and 20 mM G with 30 mM KCl (WT; N=6, *Kcnk16* L114P (L/P); N=9, and *Kcnk16* L114P (P/P); N=7). (**L–N**) Average area under the curve (AUC) analysis of normalized Ca$^{2+}$ during 7 mM G, 20 mM G, and 20 mM G+30 mM KCl stimulations. (**O**) Survival curve for *Kcnk16* L114P (P/P) mice treated with (N=6) or without (N=6) insulin (Lantus insulin glargine; 0.2 U/kg/day) starting at P0 up until death or weaning. Data are presented as mean ± SEM. Data were analyzed using Student's t-test, one-way ANOVA, and two-way ANOVA as appropriate. *p<0.05, **p<0.01, ***p<0.001, and ****p<0.0001.

The online version of this article includes the following source data and figure supplement(s) for figure 1:

**Figure supplement 1.** Generation of *Kcnk16* L114P model and assessment of neonatal glucose homeostasis and lethality.

**Figure supplement 1—source data 1.** Original file for the DNA gel analysis in *Figure 1—figure supplement 1B* (PCR confirmation of the male founder Kcnk16 L114P (L/P) mouse using Hinfl restriction digestion).

**Figure supplement 1—source data 2.** Original file for the DNA gel analysis in *Figure 1—figure supplement 1B* (PCR confirmation of the male founder Kcnk16 L114P (L/P) mouse using Hinfl restriction digestion) with bands and DNA ladder labeled.

**Figure supplement 2.** *Kcnk16* L114P male and female mice show transient neonatal hyperglycemia.

treatment (Glargine Lantus; 0.2 U/kg/day, once-daily subcutaneous injection) was able to extend the lifespan of *Kcnk16* L114P (P/P) neonates, suggesting lethality results from hyperglycemia due to inadequate insulin secretion (*Figure 1O*).

## Adult *Kcnk16* L114P mice exhibit fasting hyperglycemia and glucose intolerance

Because neonatal *Kcnk16* L114P (L/P) mice showed transient hyperglycemia (*Figure 1—figure supplement 2*), we next utilized an intraperitoneal glucose tolerance test (GTT) to determine if *Kcnk16* L114P impairs glucose homeostasis in young adulthood. Male *Kcnk16* L114P (L/P) mice developed glucose intolerance as early as 11 weeks of age in the B6;CD-1 background and 8 weeks of age in the B6 background compared to their respective control littermates (data not shown). Similarly, male B6;CD-1 *Kcnk16* L114P (L/P) mice developed impaired oral glucose tolerance by 7 weeks of age (*Figure 2—figure supplement 1*). This defect in glucose homeostasis is maintained with age in the male *Kcnk16* L114P (L/P) mice (*Figure 2A–C* and *Figure 2—figure supplements 1A–C and 2A–C*). Furthermore, male *Kcnk16* L114P (L/P) mice also exhibit fasting hyperglycemia indicating a likely non-β-cell-specific islet defect in this mouse model (*Figure 2B*). There were no body weight differences in males (*Figure 2D*). Female *Kcnk16* L114P (L/P) mice only developed moderate glucose intolerance compared to the controls (WT) in both B6 and B6;CD-1 genetic backgrounds (*Figure 2E–G*; *Figure 2—figure supplement 1D–F, 2E, F*). Moreover, the *Kcnk16* L114P mutation resulted in greater body weight gain in B6;CD-1 females compared to the WT controls (*Figure 2H*) due to increased lean mass, which might reduce glucose intolerance in *Kcnk16* L114P (L/P) females compared to males (*Figure 2—figure supplement 3A–F*). The T2D-associated SNP in *KCNK16* (rs1535500) which causes gain-of-function in TALK-1 activity has shown a significant association with increased total cholesterol in a Han Chinese population (*Permana et al., 2019*). However, an increase in TALK-1 activity due to the *Kcnk16* L114P mutation did not affect plasma and liver triglyceride and cholesterol levels in mice (*Figure 2—figure supplement 3G–J*).

## Adult *Kcnk16* L114P mice show disrupted islet hormone secretion and islet composition

We then tested if the TALK-1-L114P-mediated impairment in glucose homeostasis arises from defective islet function. Both male and female *Kcnk16* L114P (L/P) mice exhibited reduced plasma insulin levels compared to controls at 15 min and 30 min following an intraperitoneal glucose injection after a 4 hr fast (*Figure 3A and B*). Insulin sensitivity was not altered in the *Kcnk16* L114P (L/P) mice indicating that the reduction in glucose tolerance is primarily due to an islet secretion defect (*Figure 3C*).

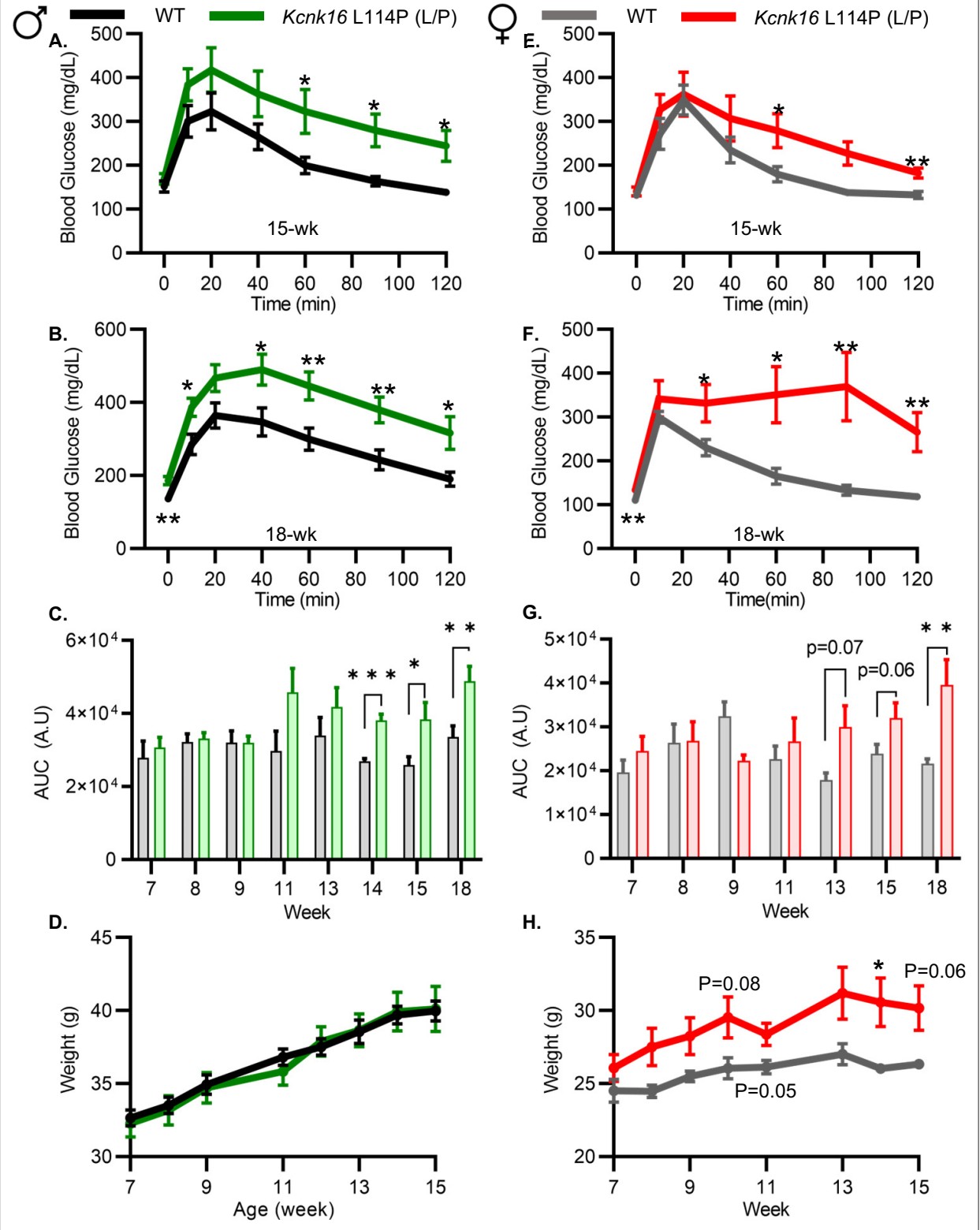

**Figure 2.** Adult *Kcnk16* L114P mice exhibit fasting hyperglycemia and glucose intolerance. (**A, B**) Intraperitoneal glucose tolerance test (i.p. GTT) performed in 15-week-old and 18-week-old male mice following a 4 hr fast in response to 2 mg/g glucose injection (WT; black; N=8–10 and *Kcnk16* L114P (L/P); green; N=9–10). (**C**) Average area under the curve (AUC) of the 2 hr GTT excursion profiles from ages 7 weeks up to 18 weeks in male mice. (**D**) Weekly body weight measurements of male WT; N=5 mice and *Kcnk16* L114P (L/P); N=5 mice. (**E, F**) I.p. GTT performed in 15-week-old and 18-week-old female mice following a 4 hr fast in response to 2 mg/g glucose injection (WT; N=9–11 and *Kcnk16* L114P (L/P); N=10–11). (**G**) Average AUC

*Figure 2 continued on next page*

*Figure 2 continued*

of the 2 hr GTT excursion profiles from ages 7 weeks up to 18 weeks in female mice. (**H**) Body weight measurements of female WT (black; N=4) and *Kcnk16* L114P (L/P; blue; N=5) mice. Data are presented as mean ± SEM. Data were analyzed using Student's t-test. *p<0.05, **p<0.01, and ***p<0.001.

The online version of this article includes the following figure supplement(s) for figure 2:

**Figure supplement 1.** Oral glucose tolerance is impaired in *Kcnk16* L114P (L/P) mice in the B6;CD-1 background.

**Figure supplement 2.** Glucose homeostasis is impaired in *Kcnk16* L114P (L/P) mice in the B6 background.

**Figure supplement 3.** Body composition measurements and assessment of plasma and liver triglycerides and total cholesterol.

This was confirmed by in vitro glucose-stimulated insulin secretion and glucose inhibition of glucagon secretion measurements (*Figure 3D and E*). Islets from *Kcnk16* L114P (L/P) mice exhibited reduced ability to secrete insulin in response to glucose and showed elevated glucagon secretion under low glucose (2 mM) and euglycemic (7 mM) conditions (*Figure 3D and E*). This indicates that the fasting hyperglycemia observed in male *Kcnk16* L114P (L/P) mice (*Figure 2B*) is likely a result of the observed increase in glucagon secretion under fasting conditions. We next investigated if impaired islet hormone secretion results from changes in islet composition. Immunostaining analyses revealed an increase in glucagon-positive area/islet and a concurrent modest reduction in insulin-positive area/islet in *Kcnk16* L114P (L/P) pancreata (*Figure 3F–J*). Together, these results indicate that *Kcnk16* L114P mutation leads to disruptions in both islet composition and secretion giving rise to a MODY-like phenotype in adult mice.

### *Kcnk16* L114P blunts β-cell glucose-stimulated electrical excitability and increases whole-cell two-pore domain K$^+$ channel currents

We used patch-clamp electrophysiology to test if the reduction in glucose-stimulated insulin secretion results from decreased β-cell electrical excitability due to a gain-of-function in TALK-1 activity in *Kcnk16* L114P mouse islets. Perforated patch-clamp recordings revealed blunting of β-cell $V_m$ depolarization and loss of action potential firing in islets from *Kcnk16* L114P (L/P) mice (*Figure 4A and B*). Because the whole islet recordings assume β-cell electrical responses in response to glucose, perforated patch-clamp recordings were also run on islet cell clusters expressing a fluorescent reporter specifically in β-cells. Similar to the whole islet recordings, *Kcnk16* L114P β-cells showed blunted glucose-stimulated $V_m$ depolarization, which did not reach a depolarized threshold to fire action potentials in 67% of β-cells (*Figure 4—figure supplement 1*). However, as 33% of *Kcnk16* L114P β-cells showed glucose-stimulated action potential firing (*Figure 4—figure supplement 1*), this indicates that these cells may also undergo Ca$^{2+}$ entry and glucose-stimulated insulin secretion. Reduced glucose-stimulated islet $V_m$ depolarization and action potential firing in *Kcnk16* L114P β-cells was in part a result of increased whole-cell two-pore domain K$^+$ channel currents at depolarized $V_m$ and thus reducing K$^+$ channel activity with high KCl depolarized these cells (*Figure 4C and D*; *Figure 4—figure supplements 1 and 2*). However, TALK-1 L114P (L/P and P/P) channels did not show a substantial increase at hyperpolarized β-cell membrane potentials under low and high glucose where these K$^+$ currents are typically active (*Figure 4—figure supplement 2*). Notably, the heterologously expressed human TALK-1 L114P channels exhibited a drastic gain-of-function (*Graff et al., 2021*); this suggests the likelihood of unknown endogenous regulator(s) of TALK-1 in β-cells. It is also likely that recording conditions lead to poor isolation of the endogenous TALK-1 L114P currents. Thus, two-pore domain K$^+$ channel currents were assessed by overexpressing mouse TALK-1 L114P channels in HEK293T cells under identical recording conditions as the human TALK-1 L114P study. Indeed, overexpression of the mouse mutant channel results in large K$^+$ currents indicative of a similar gain-of-function in mouse TALK-1 current as the human TALK-1 current due to the L114P mutation (*Figure 4E and F*).

### *Kcnk16* L114P reduces glucose- and tolbutamide-stimulated Ca$^{2+}$ entry and augments IP$_3$-induced [Ca$^{2+}$]$_{ER}$ release

β-Cell Ca$^{2+}$ entry was monitored in response to increasing concentrations of glucose in islets from WT and *Kcnk16* L114P (L/P) mice to assess if TALK-1 L114P-mediated $V_m$ hyperpolarization reduces glucose-stimulated [Ca$^{2+}$]$_c$ influx. TALK-1 L114P blunted islet glucose-stimulated [Ca$^{2+}$]$_c$ influx in mice on both B6 and B6;CD-1 genetic backgrounds compared to WT controls (*Figure 5A–F* and *Figure 5—figure supplement 1*). However, islets from male *Kcnk16* L114P (L/P) mice showed a larger reduction

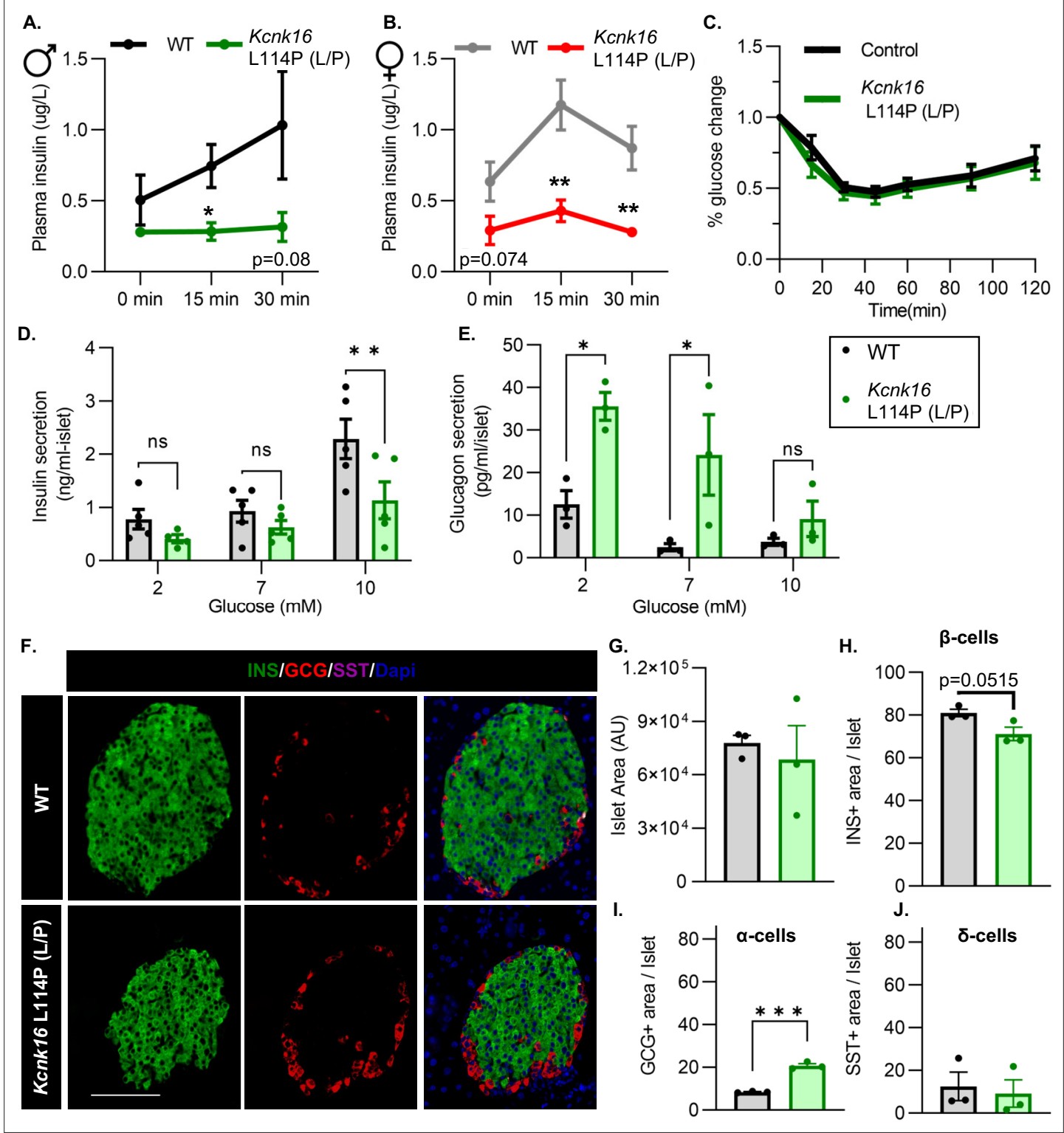

**Figure 3.** Adult *Kcnk16* L114P mice show disrupted islet hormone secretion and islet composition. (**A, B**) Plasma insulin levels in male (**A**) and female (**B**) WT and *Kcnk16* L114P (L/P) mice following a 4 hr fast at the indicated time points before and after a 2 mg/g glucose injection. (**C**) Glucose (%) change in response to an intraperitoneal (i.p.) human insulin injection (0.75 UI/kg body weight) was measured as an indicator of insulin sensitivity in WT and *Kcnk16* L114P male mice after a 4 hr fast. (**D, E**) In vitro insulin secretion (N=5 mice/genotype) and glucagon secretion (N=3 mice/genotype) from male mice at the specified glucose concentrations. (**F**) Representative immunostaining images of pancreas sections from WT and *Kcnk16* L114P (L/P) male mice (N=3/genotype) stained against insulin (INS; green), glucagon (GCG; red), somatostatin (SST; magenta), and Hoechst (blue); scale bar

*Figure 3 continued on next page*

Figure 3 continued

= 100 µm. (**G–J**) Average islet area and area of hormone staining/islet for β-cells (INS), α-cells (GCG), and δ-cells (SST). Data are presented as mean ± SEM. Data were analyzed using Student's t-test or two-way ANOVA as appropriate.*$p<0.05$, **$p<0.01$, and ***$p<0.001$.

in normalized $Ca^{2+}$ peak and normalized area under the curve responses compared to their controls than islets from female *Kcnk16* L114P (L/P) mice compared to their controls (**Figure 5A–F**). TALK-1 L114P channels did not affect the basal $Ca^{2+}$ levels at 2 mM G; however, the activity of these channels resulted in increased initial glucose-stimulated drop in $[Ca^{2+}]_c$ (termed phase 0 response) compared to controls (**Figure 5A**; **Figure 5—figure supplement 1**). This suggests a reduction in endoplasmic reticulum (ER) $Ca^{2+}$ ($[Ca^{2+}]_{ER}$) storage in *Kcnk16* L114P islets. Furthermore, *Kcnk16* L114P islets also showed a complete lack of glucose-stimulated $[Ca^{2+}]_c$ oscillations monitored at 9 mM G (**Figure 5—figure supplement 2C and D**). We then assessed if the reduction in glucose-stimulated $[Ca^{2+}]_c$ influx results from TALK-1 L114P overactivity on the β-cell plasma membrane by monitoring tolbutamide-stimulated $V_m$ depolarization-induced $[Ca^{2+}]_c$ entry. TALK-1 L114P-mediated β-cell $V_m$ hyperpolarization overrides the $V_m$ depolarization caused by $K_{ATP}$ closure by tolbutamide. Thus, *Kcnk16* L114P islets do not exhibit $[Ca^{2+}]_c$ entry in response to tolbutamide. Interestingly, islets from *Kcnk16* L114P mice showed equivalent KCl-stimulated $[Ca^{2+}]_c$ entry compared to control islets; this suggests that the loss of glucose-stimulated $[Ca^{2+}]_c$ influx in the *Kcnk16* L114P islets occurs due to enhanced $K^+$ ion flux through gain-of-function in TALK-1 activity (**Figure 5G and H**). Our previous studies showed that TALK-1 channels are functional on the ER membrane where they provide a $K^+$ countercurrent for $Ca^{2+}$ release from the ER lumen (**Vierra et al., 2017**). Thus, we tested if TALK-1 L114P regulates $[Ca^{2+}]_{ER}$ homeostasis by stimulating $G_q$-signaling using acetylcholine, which results in $IP_3$ generation and $IP_3$-induced $[Ca^{2+}]_{ER}$ release. Indeed, $IP_3$-induced $[Ca^{2+}]_{ER}$ release was enhanced in islets from *Kcnk16* L114P mice compared to controls suggesting gain-of-function TALK-1 channels are localized to the ER membrane where they facilitate increased $Ca^{2+}$ release from the ER lumen (**Figure 5I and J**).

## *Kcnk16* L114P islets exhibit altered expression of genes involved in β-cell identity, function, ion channel activity, inflammatory signaling, and extracellular matrix interaction pathways

Despite the loss of glucose-stimulated islet electrical activity and $Ca^{2+}$ entry, *Kcnk16* L114P mice do not exhibit a drastic reduction in glucose-stimulated insulin secretion and overt diabetes. Therefore, compensatory mechanisms such as the amplification pathway for insulin secretion might be altered in *Kcnk16* L114P islets (**Henquin, 2000**). Moreover, prolonged hyperglycemia (although modest in this model) would also be predicted to result in long-term gene expression changes in *Kcnk16* L114P islets that alter β-cell function. Thus, gene expression differences between WT and *Kcnk16* L114P mouse islets were quantified with bulk RNA sequencing and validated with qRT-PCR (**Figure 6A–D**). We observed increased expression of many genes which regulate $Ca^{2+}$-independent potentiation of insulin secretion such as *Adcy5*, *Creb5*, *Adcyap1*, and *Adcyap1r1*; genes that may promote $Ca^{2+}$-independent secretion from *Kcnk16* L114P islets (**Hodson et al., 2014**; **Jamen et al., 2002**). For example, *Adcy5* has been shown to amplify glucose-stimulated cAMP production, which enhances glucose-stimulated insulin secretion. As cAMP is also a critical signal for β-cell $Ca^{2+}$ entry (**Capozzi et al., 2019**; **Zaborska et al., 2022**), the impact of cAMP on *Kcnk16* L114P islet $Ca^{2+}$ handling was examined. Elevation of cAMP through activation of the GLP-1 receptor (with 200 nM liraglutide) only increased glucose-stimulated $[Ca^{2+}]_c$ oscillation frequency in WT islets and did not impact the loss of glucose-stimulated $[Ca^{2+}]_c$ entry in *Kcnk16* L114P islet (**Figure 6—figure supplement 1**). Future studies are required to determine if and how gene expression changes promote $Ca^{2+}$-independent glucose-stimulated insulin secretion in *Kcnk16* L114P islets.

Other pathways could also be impacted in *Kcnk16* L114P islets that may modify $Ca^{2+}$ handling or respond to the altered signaling such as in response to reduced glucose-stimulated $[Ca^{2+}]_c$ influx, fasting hyperglycemia, and/or impaired glucose tolerance. For example, broad changes in ion channel activity genes (e.g. *Cacna1g*, *Cacng8*, *Scn5a*, *Pkd1l1*, *Kcnk2*, *Gabrg1*, and *Fxyd3*) may contribute to altered $Ca^{2+}$ handling in *Kcnk16* L114P islets. Reduced $Ca^{2+}$ entry in *Kcnk16* L114P islets results in decreased expression of genes previously shown to be elevated in chronically depolarized $Abcc8^{-/-}$ β-cells such as *Serpina7*, *Asb11*, *Sall1*, and *Aldh1a3* (**Figure 6A**; **Stancill et al., 2017**). However,

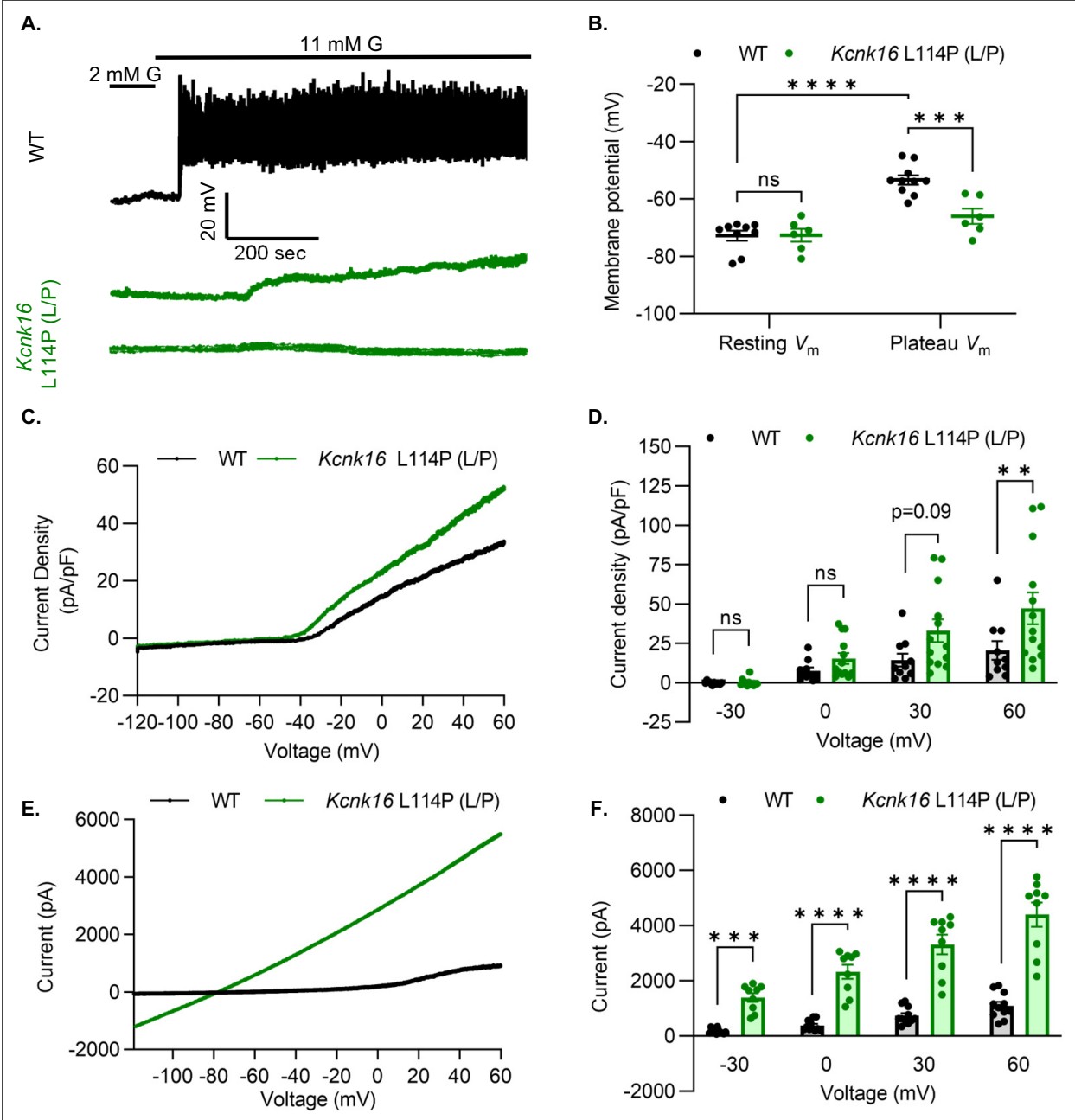

**Figure 4.** *Kcnk16* L114P blunts β-cell glucose-stimulated electrical excitability and increases whole-cell two-pore domain K⁺ channel currents. (**A**) Representative perforated patch-clamp $V_m$ recordings in response to 2 mM G and 11 mM G in islets from WT and *Kcnk16* L114P (L/P) mice (N=6–10 islets from 5 mice/genotype). (**B**) Average resting islet $V_m$ under 2 mM G and plateau islet $V_m$ at 11 mM G. (**C**) Representative whole-cell two-pore domain K⁺ channel current density (pA/pF) recorded using a voltage ramp (–120 mV to +60 mV) at 11 mM G in β-cells from WT and *Kcnk16* L114P (L/P) mice. (**D**) Average current density (pA/pF) at the specified membrane voltages during the voltage ramp recordings shown in panel C (N=9–13 cells/genotype). (**E**) Representative whole-cell two-pore domain K⁺ channel current (pA) recorded using a voltage ramp (–120 mV to +60 mV) in 11 mM G in HEK293T cells expressing either *Kcnk16* WT or *Kcnk16* L114P (L/P). (**F**) Average current (pA) at the specified membrane voltages during the voltage ramp recordings shown in panel E (N=9–11 cells/ group). Data are presented as mean ± SEM. Data were analyzed using two-way ANOVA. *p<0.05, **p<0.01, ***p<0.001, and ****p<0.0001.

The online version of this article includes the following figure supplement(s) for figure 4:

**Figure supplement 1.** *Kcnk16* L114P attenuates β-cell glucose-stimulated electrical excitability in a heterogenous manner.

**Figure supplement 2.** Two-pore domain K⁺ channel currents in β-cells from homozygous *Kcnk16* L114P (P/P) mice also exhibit a modest increase.

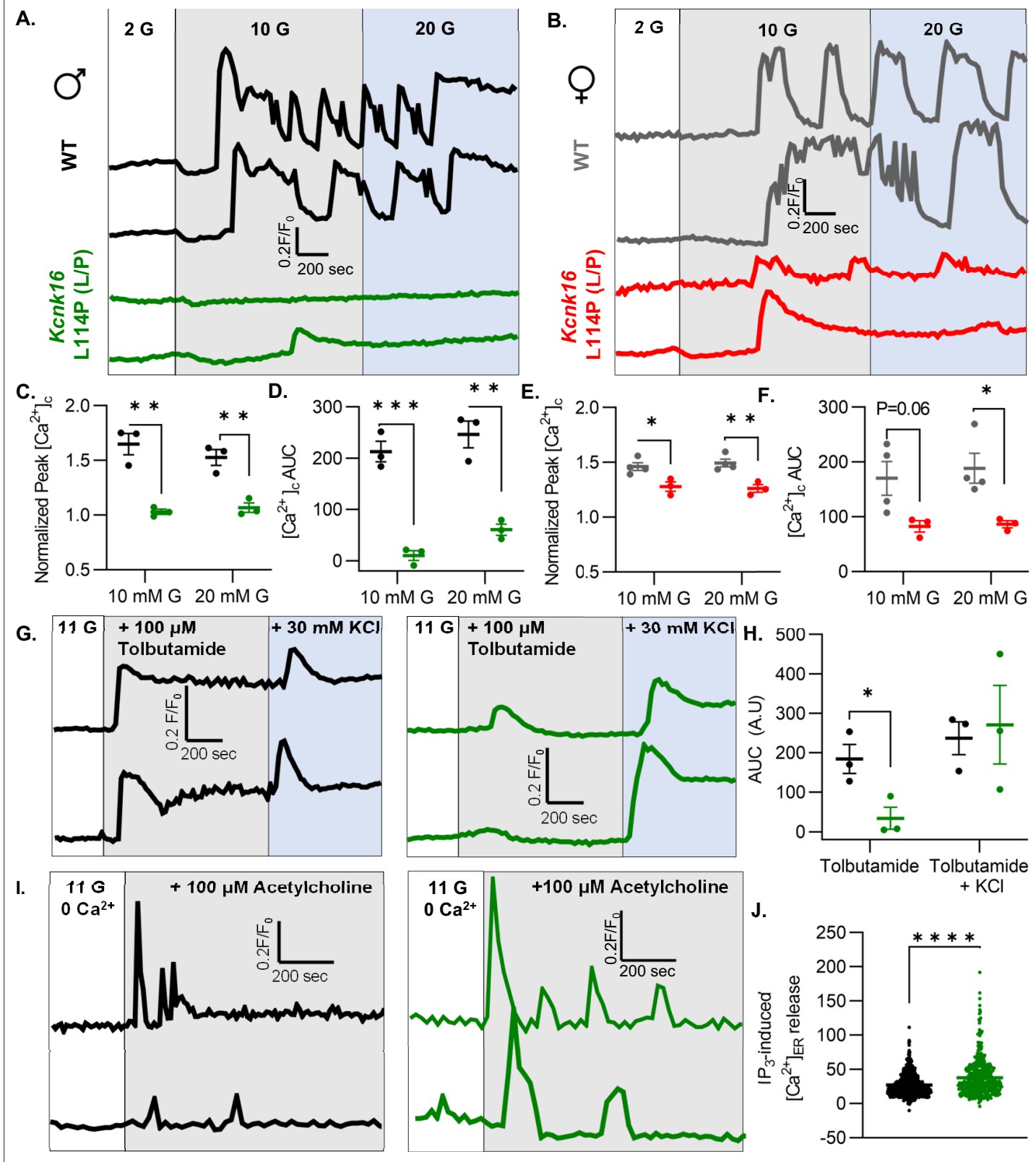

**Figure 5.** *Kcnk16* L114P reduces glucose- and tolbutamide-stimulated islet Ca²⁺ entry and augments IP₃-induced islet [Ca²⁺]ER release. (**A, B**) Representative [Ca²⁺]c traces in islets from male (**A**) and female (**B**) WT and *Kcnk16* L114P (L/P) mice in response to 2 mM G, 10 mM G, and 20 mM G. (**C–F**) Average normalized Ca²⁺ peak (C and E) and total area under the curve (AUC) (D and F) in response to the indicated glucose concentrations in islets from male and female WT and *Kcnk16* L114P (L/P) mice (N=3–4 mice/genotype). (**G**) Representative [Ca²⁺]c traces in islets from WT and

*Figure 5 continued on next page*

*Figure 5 continued*

*Kcnk16* L114P (L/P) male mice in response to 100 µM tolbutamide followed by 30 mM KCl stimulation. (**H**) Average AUC of normalized [$Ca^{2+}$]$_c$ during 100 µM tolbutamide or 100 µM tolbutamide with 30 mM KCl stimulation (N=3 mice/genotype). (**I**) Representative [$Ca^{2+}$]$_c$ traces in response to 100 µM acetylcholine in the absence of extracellular $Ca^{2+}$ in islets from male WT and *Kcnk16* L114P (L/P) mice. (**J**) Average AUC of normalized [$Ca^{2+}$]$_c$ following 100 µM acetylcholine-stimulated [$Ca^{2+}$]$_{ER}$ release (N=876 cells; WT, N=513 cells; *Kcnk16* L114P (L/P)). Data are presented as mean ± SEM. Data were analyzed using Student's t-test. *p<0.05, **p<0.01, ***p<0.001, and ****p<0.0001.

The online version of this article includes the following figure supplement(s) for figure 5:

**Figure supplement 1.** Islets from *Kcnk16* L114P (L/P) mice on the B6 background also exhibit blunted glucose-stimulated $Ca^{2+}$ entry.

**Figure supplement 2.** *Kcnk16* L114P (L/P) islets exhibit prolonged glucose-stimulated phase 0 [$Ca^{2+}$]$_{ER}$ uptake and show a complete absence of $Ca^{2+}$ oscillations.

there were no expression differences in [$Ca^{2+}$]$_{ER}$ handling genes such as *Kcnk3*, *Itpr1*, *Itpr2*, or *Iptr3* (*Figure 6—figure supplement 2*). Furthermore, several stress-associated, fibrosis-related, and inflammatory signaling pathway genes are upregulated in *Kcnk16* L114P islets likely due to prolonged hyperglycemic conditions. These include dedifferentiation markers *Sox4*, *Sox6*, *Sox9*, *Hk2*, *Vim*, and *Cd36*, extracellular matrix-interaction pathway genes *Col1a1*, *Col1a2*, *Col3a1*, *Col14a1*, *Col6a1*, *and Dcn*, and inflammatory signaling genes *Cxcl1*, *Ccl2*, *Ccl11*, *Ccl22*, *Tgfb2*, *Il33*, and *Il6* (*Roefs et al., 2017*; *Puri et al., 2013*; *Szabat et al., 2011*; *Eguchi and Nagai, 2017*; *Homo-Delarche et al., 2006*; *Hayden and Sowers, 2007*). The data show that *Kcnk16* L114P islets exhibit gene expression differences in many pathways. Taken together, this suggests that perturbed *Kcnk16* L114P islet-intrinsic (e.g. $Ca^{2+}$ handling) and -extrinsic (e.g. hyperglycemia) pathways result in direct as well as indirect disruption of islet function.

## Discussion

Gain-of-function in TALK-1 activity is associated with diabetic phenotypes, suggesting a causal role for overactive TALK-1 in islet dysfunction and diabetes progression. Utilizing a gain-of-function model of TALK-1 (L114P), this study uncovered that disrupted β-cell and α-cell function resulted in glucose intolerance in adolescent mice confirming the association of the TALK-1 L114P mutation with MODY-like diabetes. Importantly, our data also revealed that *Kcnk16* L114P mutation can cause transient neonatal diabetes. This finding suggests that transient neonatal diabetic patients with unknown genetic diagnosis should be screened for mutations in *KCNK16*. Moreover, our data provides further genetic evidence that TALK-1 is a potentially novel therapeutic target for diabetes treatment. Specifically, in mice heterozygous for *Kcnk16* L114P mutation, we observe neonatal hyperglycemia due to blunted glucose-stimulated insulin secretion, which can additionally result in neonatal death in mice homozygous for the mutant (L114P) allele of *Kcnk16*. In young adulthood, *Kcnk16* L114P causes glucose intolerance due to a reduction in glucose-stimulated insulin secretion mediated by enhanced β-cell $V_m$ hyperpolarization and reduced glucose-stimulated $Ca^{2+}$ entry. In addition to the β-cell-intrinsic defect, the *Kcnk16* L114P mutation led to an increase in α-cell area fraction in islets and elevation of glucagon secretion under fasting conditions. Together these data highlight the crucial role of TALK-1 in β-cell function and glucose homeostasis and raises the possibility of TALK-1 inhibition as a druggable target for not only *KCNK16*-associated MODY but possibly for other forms of diabetes.

β-Cell maturation and glucose responsiveness of neonatal islets rapidly develops after birth and glucose-stimulated insulin secretion is required for efficient glucose uptake, which contributes to normal growth. Following birth, a shift to intermittent feeding and elevated plasma glucose requires an increase in β-cell insulin secretion for efficient nutrient absorption, as well as a suppression of insulin release during fasting to avoid hypoglycemia (*Helman et al., 2020*). Glucose sensitivity in mouse β-cell $Ca^{2+}$ handling and insulin secretion develops over the first 4 postnatal days due to changes in β-cell metabolism, K$_{ATP}$ surface localization, and $Ca^{2+}$-dependent secretory machinery (*West et al., 2021*). Thus, mice expressing severe gain-of-function ATP-insensitive K$_{ATP}$ channels die shortly after birth due to hypoinsulinemia, severe hyperglycemia, and ketoacidosis (*Koster et al., 2000*). *Kcnk16* L114P mice show a similar phenotype, with neonatal islets showing a complete loss of glucose-stimulated [$Ca^{2+}$]$_c$ influx, a drastic reduction in glucose-stimulated insulin secretion, and severe hyperglycemia by P4. This suggests a greater impact of *Kcnk16* L114P in neonatal islets compared to adult islets. Future studies during β-cell maturation are required to determine if TALK-1 activity is greater on the

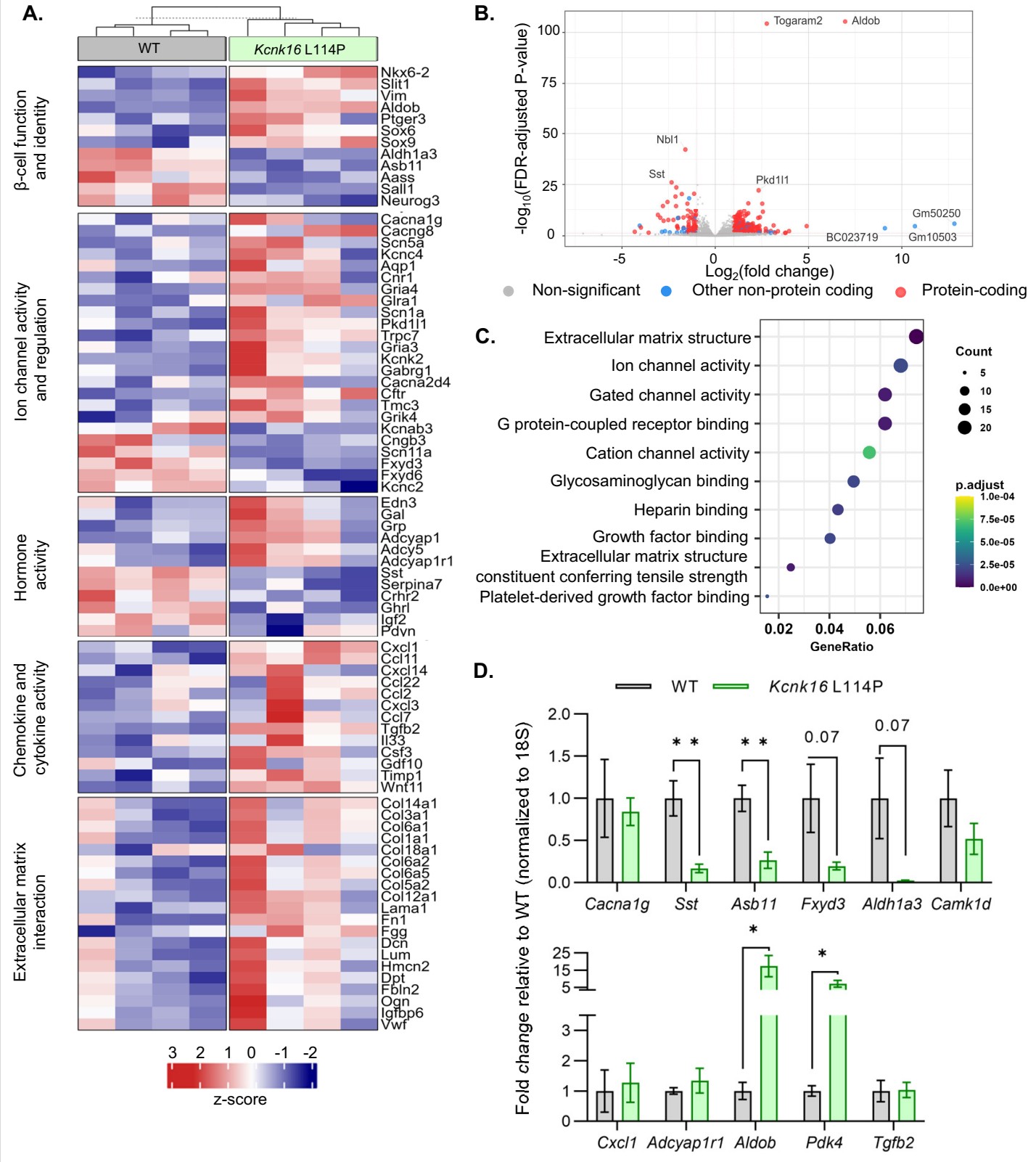

**Figure 6.** *Kcnk16* L114P islets exhibit altered expression of genes involved in β-cell identity and function, ion channel activity, hormone activity, inflammatory signaling, and extracellular matrix interaction pathways. (**A**) Heatmap of a selected gene subsets showing differential gene expression in WT and *Kcnk16* L114P islets. Normalized expression levels were scaled and centered by rows. (**B**) Volcano plot displays genes differentially expressed between WT and *Kcnk16* L114P samples. Differentially expressed genes are defined by FDR <0.05 and log2FC (≥1). (**C**) Dotplot represents the top 10

*Figure 6 continued on next page*

*Figure 6 continued*

most significantly (FDR <0.05) altered Gene Ontology (Molecular Function). GeneRatio represent (count of enriched genes)/(count of genes in the GO term). The color represents FDR-adjusted p-values and the size of the dot represents the number of genes that are significant from the experimental dataset. (D) qRT-PCR validation of the gene expression differences in WT and *Kcnk16* L114P samples for the selected genes observed through bulk RNA sequencing. *p<0.05, and **p<0.01.

The online version of this article includes the following figure supplement(s) for figure 6:

**Figure supplement 1.** Liraglutide increased $Ca^{2+}$ oscillation frequency in WT islets but does not impact *Kcnk16* L114P (L/P) mouse islet $Ca^{2+}$ handling.

**Figure supplement 2.** RNA sequencing showed equivalent expression of a subset of genes encoding proteins known to control $[Ca^{2+}]_{ER}$ in WT and *Kcnk16* L114P (L/P) male islets.

plasma membrane and/or ER membrane in early neonatal compared with adult β-cells. Interestingly, *Kcnk16* L114P-mediated susceptibility to neonatal lethality was dependent on genetic diversity in mouse strains. Heterozygous *Kcnk16* L114P caused almost complete neonatal lethality in the B6 background but only hyperglycemia in the B6:CD-1 background, whereas homozygous *Kcnk16* L114P led to neonatal lethality in the B6:CD-1 mice. The mechanism for the enhanced ability of heterozygous *Kcnk16* L114P B6:CD-1 mice to survive remains to be determined. However, this has previously been reported for other knockout mouse models (e.g. *Ankfy1* and *Ywhaz*), that are only viable as adults on a mixed genetic background (*Weng et al., 2016*; *Yang et al., 2017*). Although the neonatal glycemic data from the *Kcnk16* L114P families does not exist, it may be that neonatal lethality is not observed in affected individuals. However, it is interesting to note that all currently identified individuals carrying the *Kcnk16* L114P mutation are females (N=8) (*Graff et al., 2021*; *Katsuyuki Matsui et al., 2023*). The importance of hypoinsulinemia causing neonatal diabetes and lethality in TALK-1 L114P mice was confirmed by insulin treatment which extended their lifespan. This resembles other monogenic forms of neonatal diabetes that require exogenous insulin treatment for survival (*Polak and Cavé, 2007*). Indeed, it has been well established that insulin signaling is required for neonatal survival; for example, a similar neonatal lethality phenotype was observed in mice without insulin (*Ins1/Ins2* double knockout) where death results within 2 days as well as in mice without insulin receptors (*Insr$^{-/-}$*) where death results by P3 (*Accili et al., 1996*; *Duvillié et al., 1997*). Future studies are required to determine if TALK-1 gain-of-function mutations result in human transient neonatal diabetes followed by a MODY phenotype. Other monogenic diabetes mutations such as in genes encoding $K_{ATP}$ channels result in transient neonatal diabetes with diabetes reemergence later in life (*Flanagan et al., 2007*).

In adolescence, the timeline of development of MODY-like diabetes in the *Kcnk16* L114P (L/P) model is consistent with data from MODY patients. Similar to the timeline of disease progression in other MODY mouse models, *Kcnk16* L114P mice developed glucose intolerance during adolescence (~8 weeks in the B6 strain). The onset and severity of glucose intolerance in *Kcnk16* L114P mice also recapitulates the data from *KCNK16* L114P MODY family probands, who were diagnosed at 11 and 15 years of age and displayed an abnormal oral GTT (blood glucose: 19 and 19.6 mM, 2 hr after 75 g oral glucose bolus, respectively). Disease severity is more prominent in the inbred B6 strain compared to the mixed B6:CD-1 strain likely owing to the outbred characteristics and genetic diversity of the CD-1 strain, which recapitulates the diverse nature of MODY manifestation observed in human patients. As the Japanese *KCNK16* MODY family showed a greater insulin requirement compared to the patients from Australian *KCNK16* MODY family, diversity in diabetes phenotypes likely occurs with *KCNK16* mutations (*Graff et al., 2021*; *Katsuyuki Matsui et al., 2023*). Interestingly, male *Kcnk16* L114P (L/P) mice exhibit more severe impairment in glucose homeostasis compared to female *Kcnk16* L114P (L/P) mice. A variety of factors could explain the observed sex differences, including the female hormone 17β-estradiol (E2) which is critical for protection against glucolipotoxicity and oxidative stress (*Gannon et al., 2018*). Furthermore, older women have higher insulin levels in relation to relative insulin demand compared to men (*Basu et al., 2017*). Glucose-stimulated $[Ca^{2+}]_c$ influx in islets from female *Kcnk16* L114P mice is significantly greater than islets from male *Kcnk16* L114P mice, which would be predicted to lead to greater insulin secretion and lesser impairment in glucose tolerance in female *Kcnk16* L114P mice. Sexual dimorphism observed in glucose-stimulated $[Ca^{2+}]_c$ influx could be mediated by differences in $Ca^{2+}$ handling or differences in TALK-1 function. For example, under stress female human T2D islets maintain greater insulin secretion compared to males (*Brownrigg et al., 2023*). Although greater β-cell *Kcnk16* expression could also be responsible for the more significant impairment in glucose intolerance observed in *Kcnk16* L114P male mice, there

is only a trend for greater *Kcnk16* expression in sorted male β-cells (average RPKM 6296.25±953.84) compared to sorted female β-cells (5148.25±1013.22); this is similar in β-cells from high-fat diet (HFD)-treated mice (average RPKM 8020.75±1944.41 for males, and average RPKM 7551±2952.70 for females) (*Brownrigg et al., 2023*). Whether sex differences exist in humans with TALK-1 L114P mutation remains to be determined as all affected individuals in both *KCNK16*-MODY (p. TALK-1 L114P) families were females. Neonatal lethality was more penetrant in male than female *Kcnk16* L114P (P/P) mice, thus, it will also be important to determine if *KCNK16*-MODY patients show neonatal phenotypes and display sexual dimorphism.

Similar to the proband from the human *KCNK16* L114P MODY family who showed elevated fasting blood glucose (~7 mM), *Kcnk16* L114P mice also exhibit fasting hyperglycemia. While this was predicted to be due to decreased insulin secretion, *Kcnk16* L114P islets showed equivalent insulin secretion under euglycemic conditions. However, these islets exhibited a significant elevation in fasting glucagon secretion which likely contributes to the fasting hyperglycemia. If TALK-1 L114P channels were expressed in α-cells, it would result in inhibition of α-cell $Ca^{2+}$ entry and glucagon secretion which supports previously described lack of TALK-1 protein expression in α-cells (*Vierra et al., 2018*). This suggests that hyperglucagonemia in TALK-1 L114P islets is likely due to loss of inhibitory paracrine signaling. As insulin secretion does not change under fasting and euglycemic conditions, hyperglucagonemia might primarily be mediated by reduced somatostatin secretion. This is supported by a significant decrease in *Sst* expression in *Kcnk16* L114P islets (*Figure 6*), which is likely due to reduced δ-cell secretion. Also, our previous data in global TALK-1 KO mice show higher somatostatin secretion and lower glucagon secretion, thus δ-cell TALK-1 L114P would be predicted to limit $Ca^{2+}$ influx and somatostatin secretion (*Vierra et al., 2018*). Intriguingly, *Kcnk16* L114P islets additionally exhibit increased α-cell area fraction and a concurrent modest reduction in β-cell area fraction compared to control islets. These changes in islet composition is consistent with both T1D and T2D data showing increased α-cell:β-cell ratio (*Li et al., 2000*; *Yoon et al., 2003*), which show elevated islet glucagon secretion. α-Cell hyperplasia may result from increased activity/secretion, which is supported by other mediators of α-cell secretion (e.g. amino acids) that cause hyperplasia as well (*Dean, 2020*). However, loss of somatostatin also elevates α-cell secretion without altering α-cell mass (*Hauge-Evans et al., 2009*). Moreover, the increased somatostatin and insulin secretion only result in reduced α-cell secretion without altering α-cell number. Thus, the exact mechanism of how TALK-1 L114P mediates increased α-cell number remains to be determined. Taken together, the *KcnK16* L114P mouse model shows disrupted glucagon and insulin secretion leading to fasting hyperglycemia and glucose intolerance, which provides confirmation that TALK-1 gain-of-function mutations likely cause MODY.

One obstacle in determining how MODY-associated mutations result in β-cell dysfunction is the limited availability of primary islet tissue from MODY families. Due to this, the initial assessment of MODY-associated human TALK-1 L114P was performed in overexpression systems, which resulted in a drastic gain-of-function. Similarly, overexpression of mouse *Kcnk16* L114P leads to a substantial gain-of-function in TALK-1 activity (7.96-fold at –30 mV and 6.13-fold at 0 mV compared to TALK-1 WT). However, surprisingly β-cells from *Kcnk16* L114P (L/P and P/P) mice showed only a modest gain-of-function in two-pore domain $K^+$ channel currents. While the slight increase in β-cell $K^+$ conductance from *Kcnk16* L114P mice would be predicted to partially alter islet excitability, these islets exhibit a complete loss of glucose-stimulated $V_m$ depolarization. This suggests that endogenous TALK-1 L114P polarizes plasma membrane potential, which is further supported by the robust KCl-induced $V_m$ depolarization and $Ca^{2+}$ entry observed in these islets. Because KCl shifts the reversal potential of $K^+$ channels to a more depolarized $V_m$, the constant $V_m$ hyperpolarization in TALK-1 L114P β-cells likely results from increased $K^+$ conductance through these channels. Yet, $K^+$ conductance in TALK-1 L114P β-cells was not significantly different at the membrane potentials of these cells under either low or high glucose conditions. The recording conditions may lead to poor isolation of the endogenous TALK-1 L114P currents; however, overexpression of this mutant channel results in large $K^+$ currents under identical recording conditions. The $K^+$ conductance differences between heterologously versus endogenously expressed TALK-1 L114P channels points toward unidentified regulators of β-cell TALK-1 activity which could include endogenous ligands, protein interactions, and cellular localization of the channel. This is presumably not due to changes in TALK-1 protein levels because *Kcnk16* mRNA expression was not altered in control and *Kcnk16* (c. 337T>C) islets. The subtle increase in β-cell $K^+$ conductance

correlates with a modest MODY-like phenotype and likely allows for incomplete suppression of β-cell function. This is also observed in K$_{ATP}$-MODY, where channel activity is only modestly increased (*Yori-fuji et al., 2005*). Importantly, our study suggests that drastic TALK-1 gain-of-function mutations only lead to modest β-cell K$^+$ conductance, which may explain why both families with TALK-1 MODY carry the same pore domain mutation (L114P) in TALK-1. Additionally, it is likely that other less-severe gain-of-function mutations in TALK-1 (e.g. A277E) result in a milder phenotype such as T2D. Another possibility is that because TALK-1 channels are functional not only on the plasma membrane but also the ER membrane (*Vierra et al., 2017*), some of the TALK-1 L114P-mediated reduction in glucose-stimulated [Ca$^{2+}$]$_c$ influx could be due to altered [Ca$^{2+}$]$_{ER}$ handling. Indeed, glucose-stimulated [Ca$^{2+}$]$_c$ influx has previously been shown to be dependent in part on [Ca$^{2+}$]$_{ER}$ release and TALK-1 L114P increases β-cell [Ca$^{2+}$]$_{ER}$ release in response to muscarinic stimulation (*Postic et al., 2023*). Thus, future studies are required to determine the contributions of ER membrane localized versus plasma membrane localized TALK-1 L114P to altered glucose-stimulated [Ca$^{2+}$]$_c$ influx. Moreover, it will be important to establish the contributions of endogenous modulators of TALK-1 channels (WT and L114P) and how they contribute to β-cell dysfunction.

Suppression of glucose-stimulated electrical activity and [Ca$^{2+}$]$_c$ influx in *Kcnk16* L114P islets would be predicted to cause a greater reduction of glucose-stimulated insulin secretion than that observed. However, a few β-cells from *Kcnk16* L114P mice show modest $V_m$ depolarization; this resulted in a slight increase in [Ca$^{2+}$]$_c$ in response to glucose in a small subset of *Kcnk16* L114P islets, which could result in some glucose-stimulated insulin secretion. Interestingly, *Kcnk16* L114P islets also show reduced expression of *Fxyd3*, which encodes the auxiliary subunit of Na$^+$/K$^+$-ATPase and is a known negative regulator of glucose-stimulated insulin secretion in diabetic mice and humans (*Vallois et al., 2014*). Other interesting gene expression differences that may increase glucose-stimulated insulin secretion in *Kcnk16* L114P islets include elevated expression (2.13-fold) of *Adcy5*, a Ca$^{2+}$-independent amplification pathway gene (*Hodson et al., 2014*; *Kalwat and Cobb, 2017*). In humans, *Adcy5* depletion impairs glucose-dependent elevation of cAMP and associated insulin secretion, thus elevation in *Adcy5* expression would be predicted to increase glucose-stimulated insulin secretion via cAMP signaling (*Hodson et al., 2014*). Expression levels of other cAMP-dependent pathway genes were also elevated in the *Kcnk16* L114P islets including *Creb5, Adcyap1,* and *Adcyap1r1* (*Jamen et al., 2002*). Moreover, *Slit1* and *Srgap3*, part of the SLIT-ROBO signaling which enhances glucose-stimulated insulin secretion, were upregulated (*Yang et al., 2013*). SLIT-ROBO signaling regulates not only Ca$^{2+}$ handling but also actin remodeling and thus, Ca$^{2+}$-independent signaling pathways. These gene expression changes suggest an increase in glucose-stimulated [Ca$^{2+}$]$_c$ influx-independent mechanism(s) of insulin secretion may compensate for TALK-1 L114P-mediated loss of β-cell electrical activity and Ca$^{2+}$ entry.

Chronic hyperglycemia in diabetic patients and rodents results in islet dysfunction and destruction, thus hyperglycemia observed in the *Kcnk16* L114P mice could exacerbate β-cell failure. Glucotoxicity results in numerous islet transcriptome changes that in part contribute to dysfunction. Similar to islets from diabetic patients, *Kcnk16* L114P islets exhibit an increase in *Aldob* (17.45-fold) and *Nnat* (1.95-fold) expression (*Gerst et al., 2018*; *Millership et al., 2018*). Additionally, *Pdk4* expression, a marker for the shift from utilization of glucose to fatty acids as the primary fuel source, is higher in *Kcnk16* L114P islets (2.78-fold). Elevated islet PDK4 expression is also observed in patients with T2D and in animals on an HFD (*Eguchi and Nagai, 2017*). Although hyperglycemia would be predicted to negatively impact *Kcnk16* L114P islet function, no overt changes in β-cell mass were observed in these mice. This differs from mice with islets expressing K$_{ATP}$ gain-of-function mutation, which show loss of β-cell mass (*Shyr et al., 2019*). *Kcnk16* L114P may not cause a significant β-cell destruction because these mice show sufficient glucose-stimulated insulin secretion to prevent overt diabetes.

In summary, we showed that the MODY-associated TALK-1 L114P mutation elevates α-cell glucagon secretion under fasting and euglycemic conditions and blunts glucose-stimulated β-cell electrical activity and Ca$^{2+}$ entry leading to reduced insulin secretion. Together, elevated glucagon impairs fasting glycemia and reduced glucose-stimulated insulin secretion increases post-prandial glucose levels in adults. This phenotype was more prominent in the male *Kcnk16* L114P mice compared to the female *Kcnk16* L114P mice raising the question whether sex differences translate in humans carrying this mutation. Surprisingly, the TALK-1 L114P mutation also resulted in severe transient neonatal diabetes which was lethal in the C57Bl/6J genetic background. Thus, these data hold potential clinical utility in that neonatal diabetes patients with unknown genetic linkage should be screened for

mutations in *KCNK16*. Together, these data strengthen the rationale for designing TALK-1 inhibitors for use as a therapeutic modality to treat diabetes.

Limitations of the study: (1) This study does not establish how TALK-1 channels on the ER membrane influence glucose-stimulated $[Ca^{2+}]_c$ influx. Future investigations are required to determine how *Kcnk16* L114P modulation of β-cell $[Ca^{2+}]_{ER}$ contributes to blunted glucose-stimulated $[Ca^{2+}]_c$ influx and insulin secretion. (2) The *Kcnk16* L114P mouse model created for this study replicates the MODY mutation in humans, however, due to TALK-1 expression in cells other than β-cells (e.g. δ-cells) it is difficult to determine the exact cellular contributions of *Kcnk16* L114P to the MODY phenotype. (3) This study does not establish why TALK-1 L114P channel currents either from *Kcnk16* L114P (L/P) or (P/P) primary β-cells show such low conductance compared to heterologously expressed TALK-1 L114P channels.

## Methods

### Chemicals and reagents

All research materials were purchased from Thermo Fisher (Waltham, MA, USA) or Sigma-Aldrich (St. Louis, MO, USA) unless otherwise specified.

### Mouse model generation and ethical approval

Neonatal mice used for the studies were P0- to P5-old, and adult mice used for the studies were 6- to 26-week-old, age- and gender- matched, bred in-house on a C57BL/6J or mixed C57BL/6J:CD-1 (ICR) background. Animals were handled in compliance with guidelines approved by the Vanderbilt University Animal Care and Use Committee protocols (#M2200007-00). C57BL/6J.*Kcnk16*L114P mice (Kcnk16<em1Djaco>; MGI:7486559) were produced by the Vanderbilt Genome Editing Resource (Vanderbilt University, Nashville, TN, USA). Ribonucleoprotein complexes comprising chemically modified ctRNA (crRNA+tracrRNA) (50 ng/μl) and enhanced specificity SpCas9 protein (100 ng/μl), together with a 180-nucleotide single-stranded DNA (ssDNA) donor containing the *Kcnk16* L114P mutation (50 ng/μl), were obtained from MilliporeSigma (Burlington, MA, USA). These components were diluted in 10 mM Tris, 0.1 mM EDTA, pH 7.6, sourced from Teknova (Half Moon Bay, CA, USA), and administered via pronuclear injection into C57BL/6J embryos acquired from mice from Jackson Labs (Bar Harbor, ME, USA). crRNA sequence: 5'CCCTGCAGGTTATGGAAACC. 180-Nucleotide ssDNA sequence: 5' CTAGAGCTGGTGGTTGGGGGTGGGAGCCAGTTCTGGGCTCTCTTTTCCCCGCATCT GCACACTCCCTTGCCCTGCAGGTTATGGGAATCCAGCCCCCAGCACGGAGGCAGGGCAGGTCTT CTGTGTCTTCTATGCTCTGATGGGGATCCCACTCAATGTGGTCTTCCTCAACCATCTGGG. Mosaic F0 animals were screened for the L114P point mutation by standard PCR followed by a restriction fragment length polymorphism assay for a de novo HinfI site incorporated with silent mutations into the ssDNA. Animals carrying the desired mutation were confirmed by Sanger sequencing. Founder *Kcnk16* L114P (L/P) mouse was backcrossed onto the C57Bl/6J (B6) strain for two generations to obtain mice used for all studies performed on the B6 background. Additionally, B6 *Kcnk16* L114P (L/P) male mice were crossed with the CD-1 (ICR) strain to obtain F1 mice on a hybrid B6;CD-1 background to reduce the incidence of neonatal lethality. Crossings of heterozygous F1 B6;CD-1 *Kcnk16* L114P (L/P) mice were used for generating homozygous B6;CD-1 *Kcnk16* L114P (P/P) mice. For all the studies, littermates expressing the wildtype *Kcnk16* allele were used as controls (WT).

### Immunofluorescence

Mouse pancreata were fixed in 4% paraformaldehyde and embedded with paraffin. Rehydrated 5 μm sections were stained with primary antibodies against insulin (dilution 1:1000; Dako, Santa Clara, CA, USA), somatostatin (dilution 1:300, GeneTex., Irvine, CA, USA), and glucagon (dilution 1:100; Abcam, Cambridge, UK) followed by secondary antibodies (dilution 1:500; anti-guinea pig Alexa Fluor 488, dilution 1:500; anti-mouse, Alexa Fluor 647, and dilution 1:500; anti-rabbit Alexa Fluor 546) as previously described (*Vierra et al., 2015*). Sections were imaged either with a Nikon Eclipse TE2000-U microscope or fluorescent ScanScope (Aperio).

### Islet isolation

Islets from neonatal mice were isolated on P4 using the protocol described by *Huang and Gu, 2024*. Briefly, pancreata were isolated and broken into ~2 mm pieces and digested in 200 μl collagenase P

(Roche, Basel, Switzerland). For digestion, the tube was left in a 37°C incubator for up to 15 min and inverted two times every min. The lysate was spun at 500 × $g$ for 10 s followed by three washes in RPMI, after which the islets were handpicked using a brightfield microscope.

Islets from adult mouse pancreata were isolated by collagenase P digestion and density gradient centrifugation as previously described (*Vierra et al., 2015*). Following isolation, islets were either dispersed into clusters of cells or single cells with trituration in 0.005% trypsin or maintained as whole islets. Cells were cultured in RPMI 1640 supplemented with 15% FBS, 100 IU/ml penicillin, 100 mg/ml streptomycin, and 5.5 mM glucose (RPMI) in a humidified incubator at 37°C with an atmosphere of 95% air and 5% $CO_2$.

## Whole-cell two-pore domain K$^+$ channel currents

TALK-1 L114P currents were monitored using the whole-cell patch-clamp technique using an Axopatch 200B amplifier with pCLAMP10 software. Digidata 1440 was used to digitize currents that were low-pass-filtered at 1 kHz and sampled at 10 kHz. Cells were washed with the extracellular buffer (modified Krebs-Ringer-HEPES buffer [KRHB]) containing (mM) 119.0 NaCl, 2.0 $CaCl_2$, 4.7 KCl, 25.0 HEPES, 1.2 $MgSO_4$, 1.2 $KH_2PO_4$, and 11 mM glucose (pH 7.4 with NaOH). For isolation of two-pore domain K$^+$ channel currents, $K_{ATP}$ channels were blocked with 100 μM tolbutamide, voltage-gated K$^+$ channels were blocked with 10 mM tetraethylammonium (*Vierra et al., 2018*; *Vierra et al., 2017*). Patch electrodes (3–5 MΩ) were backfilled with intracellular solution (IC) containing (mM) 140.0 KCl, 1.0 $MgCl_2$, 10.0 EGTA, 10.0 HEPES, and 4.0 Mg-ATP (pH 7.25 with KOH). β-Cell $V_m$ was ramped from –120 mV to +60 mV from a holding potential of –80 mV to generate two-pore domain K$^+$ channel currents. Currents were measured in single β-cells from WT or *Kcnk16* L114P (L/P) mice, or in HEK293FT cells (Catalog no. R70007, Invitrogen, Waltham, MA, USA) expressing either *Kcnk16* WT or mouse *Kcnk16* L114P channels. The whole-cell currents were analyzed using ClampFit (Molecular Devices) and Excel (Microsoft Corp., Redmond, WA, USA).

For two-pore domain K$^+$ channel recordings in HEK293FT cells, the cells were grown to ~80% confluency in Dulbecco's Modified Eagle Media (DMEM) GlutaMax-I (Thermo Fisher Scientific) supplemented with 10% fetal bovine serum (FBS, Gibco), 100 IU/ml penicillin (Gibco), and 100 mg/ml streptomycin (Gibco) at 37°C, 5% $CO_2$ in 100 mm tissue culture dishes. Cells were transfected with either pLV-CMV-m*Kcnk16*:P2A:EGFP or pLV-CMV-m*Kcnk16* L114P:P2A:EGFP plasmids using Lipofectamine 3000 and P3000 (Thermo Fisher Scientific) in antibiotic-free Opti-MEM I Reduced Serum Medium as per the manufacturer's protocol. Two-pore domain K$^+$ channel currents were only recorded from EGFP-positive cells.

## β-Cell $V_m$ recordings

β-Cell $V_m$ was recorded by the perforated patch-clamp technique using an Axopatch 200B amplifier with pCLAMP10 software on whole islets or islet clusters (containing 5–10 cells transduced with an adenoviral construct expressing GCaMP6s from a rat insulin promoter; *Dickerson et al., 2022*). Cells or islets were washed with KRHB with (mM) 119.0 NaCl, 2.0 $CaCl_2$, 4.7 KCl, 25.0 HEPES, 1.2 $MgSO_4$, 1.2 $KH_2PO_4$ (adjusted to pH 7.4 with NaOH) supplemented with 2 mM glucose and incubated in KRHB for 30 min at 37°C, 5% $CO_2$. Patch electrodes (3–5 MΩ) were backfilled with IC containing (mM) 140.0 KCl, 1.0 $MgCl_2$, and 5.0 HEPES (adjusted to pH 7.2 with KOH) supplemented with 20 μg/ml amphotericin B. For the islet clusters only β-cells expressing GCaMP6s were recorded. Islets and islet clusters were perifused with KRHB supplemented with 2 mM glucose followed by KRHB with 10 mM glucose for monitoring $V_m$ changes. β-Cell $V_m$ recordings were analyzed using ClampFit (Molecular Devices), Excel (Microsoft Corp., Redmond, WA, USA), and GraphPad Prism 8 (GraphPad Software Inc).

## Intracellular Ca$^{2+}$ imaging

On the day of experiment, islets were incubated for 30 min in RPMI supplemented with 2 μM Fura-2, AM (Molecular Probes) and 2 mM glucose. Fura-2, AM fluorescence (Ratio 340Ex/380Ex-535Em; $F_{340}$/$F_{380}$) was measured every 5 s as an indicator of intracellular Ca$^{2+}$ using a Nikon Eclipse Ti2 microscope equipped with a Photometrics Prime 95B 25 mm sCMOS Camera (*Zaborska et al., 2020*). For [Ca$^{2+}$]$_c$ measurements, β-cell glucose-stimulated Ca$^{2+}$ influx was monitored in KRHB supplemented with the glucose concentrations specified in the figures. For IP$_3$-induced [Ca$^{2+}$]$_{ER}$ release measurements, islets were perifused in KRHB buffer containing 11 mM glucose, 100 μM diazoxide, without extracellular

$Ca^{2+}$. Fura-2, AM fluorescence was monitored as an indicator of $IP_3$-mediated $[Ca^{2+}]_{ER}$ release upon stimulation of muscarinic receptor signaling by 100 µM acetylcholine. For all measurements, the cells were perifused at a flow rate of 2 ml/min. Ex; excitation wavelength (nm), Em; emission wavelength (nm).

### Glucose homeostasis

Chow-diet fed male and female mice underwent GTT and insulin tolerance test (ITT) as previously described (*Vierra et al., 2015*). Briefly, mice were fasted for 4 hr and then 2 mg dextrose/g body weight was administered with either intraperitoneal injection or oral gavage for GTTs, or 0.75 UI human recombinant insulin/kg body weight for ITTs (Catalog no. 12585014, Gibco). Tail glucose measurements were then taken at the indicated time points in the figures to measure glucose clearance.

### Body composition, tissue and plasma triglyceride, and cholesterol measurements

Measurement of lean tissue, fat, and fluid in living mice was performed using Bruker's minispec Body Composition Analyzer. Plasma samples and livers were collected from ad lib fed mice for triglyceride and cholesterol measurements. Total cholesterol and triglycerides were measured using standard enzymatic assays by the Vanderbilt University Medical Center Lipid Core.

### Plasma insulin, in vitro insulin, and glucagon secretion assays

Plasma insulin from the neonates was measured on P4 using Ultrasensitive Mouse Insulin ELISA kits (Catalog no. 10-1249-01). In adult mice, plasma insulin measurements were conducted in mice fasted for ~4 hr followed by intraperitoneal injection of 2 mg dextrose/g body weight at 0, 15, and 30 min post injection. Tail blood samples were collected in Microvette CB 300 K2 EDTA tubes (Catalog no. 16.444.100, Sarstedt) at the indicated time points and plasma insulin were measured using mouse ultrasensitive insulin ELISA kits (Mercodia Inc, Sweden). For in vitro insulin and glucagon secretion assays, islets were isolated from mice fed a standard chow diet and were incubated overnight in RPMI supplemented with 0.5 mg/ml BSA. On the following day, islets were equilibrated in DMEM containing 0.5 mg/ml BSA, 0.5 mM $CaCl_2$ and 10.0 mM HEPES (DMEM*) supplemented with 10% FBS and 5.5 mM glucose for 1 hr at 37°C, 5% $CO_2$. 20 islets/well were picked into 400 µl DMEM* without FBS at glucose concentrations specified in the figures in 24-well plate(s) and insulin or glucagon secretion was measured over 1 hr at 37°C and stored at –20°C until analysis. Insulin secretion was measured using mouse insulin ELISA kits (Catalog no. 10-1247-01, Mercodia Inc, Sweden) and glucagon secretion was measured using mouse glucagon ELISA kits (Catalog no. 10-1281-01, Mercodia Inc, Sweden).

### Bulk RNA sequencing

RNA was isolated from islets from ~15-week-old male mice (WT and *Kcnk16* L114P (L/P)) using Maxwell 16 LEV simplyRNA Purification Kits (Catalog no. AS1280, Promega, USA). RNA integrity was analyzed using an Agilent 2100 Bioanalyzer, and only those samples with an RNA integrity number of seven or above were used. An Illumina NovaSeq 6000 instrument was used to produce paired-end, 150-nucleotide reads for each RNA sample. Paired-end RNA sequencing reads (150 bp long) were trimmed and filtered for quality using Trimgalore v0.6.7 (*Felix Krueger et al., 2023*). Trimmed reads were aligned and counted using Spliced Transcripts Alignment to a Reference (STAR) (*Dobin et al., 2013*) v2.7.9a with the –quantMode GeneCounts parameter against the mm10 mouse genome and GENCODE comprehensive gene annotations (Release M23). Per sample, the number of mapped reads ranged from 52 to 462 million. DESeq2 package v1.34.0 (*Love et al., 2014*) was used to perform normalization and downstream differential expression. Features counted fewer than five times across at least three samples were removed. Freezing condition of the samples and sequencing batch was included as batch factor in DESeq2 design to increase the sensitivity for finding differences between *Kcnk16* L114P v/s WT samples and confirmed with principle component analysis (*Figure 6—figure supplement 2B*). Gene enrichment analysis implemented from Gene Ontology (GO) was applied using the clusterProfiler v4.2.2 package in R. Annotated gene sets GO was sourced from Genome wide annotation for Mouse (*Carlson, 2019*). For GO, genes significantly up- or downregulated in different conditions were used as input. False discovery rate-adjusted p-value <0.05 and log2 fold change >1 was used to define differentially expressed genes.

## Quantitative PCR

RNA was isolated from islets from 15-week-old, chow diet fed mice using Maxwell 16 LEV simplyRNA Purification Kits (Catalog no. AS1280, Promega, USA). Reverse transcription was performed using a SuperScript IV First-Strand Synthesis System (Catalog no. 18091050, Invitrogen, Waltham, MA, USA). 20 ng cDNA was used for real-time qPCRs with KAPA SYBR FAST qPCR Kit (Catalog no. KK4618, Roche, Basel, Switzerland) using a CFX Opus Real-Time PCR System (Bio-Rad Laboratories). Primers used for qRT-PCR are listed in *Supplementary file 1*.

## Statistical analysis

Functional data were analyzed using Axon Clampfit (Molecular Devices), GraphPad Prism 8 (GraphPad Software Inc), or Excel (Microsoft Corp., Redmond, WA, USA) and presented as mean ± standard error (SE) for the specified number of samples (N). Statistical significance was determined using two-tailed t-tests, one-way ANOVA, or two-way ANOVA as appropriate. p-Value ≤0.05 was considered statistically significant.

## Acknowledgements

The *Kcnk16* L114P mouse model was developed by the Vanderbilt Genome Editing Resource Core (RRID:SCR018826) which is supported by the Diabetes Research and Training Center Grant (DK020593), and the Cancer Center Support Grant (CA68485), and the Vanderbilt Center for Stem Cell Biology. Paraffin-embedded pancreata were prepared and processed by The Vanderbilt Translational Pathology Shared Resource, which was supported by the Cancer Center Support Grant CA68485. Immunostaining slide scanning was performed using the Islet and Pancreas Analysis (IPA) Core supported by the Vanderbilt Diabetes Research Center (DRTC; NIH grant DK20593). Analysis of the bulk RNA sequencing data was performed by Creative Data Solutions, part of the Vanderbilt Center for Stem Cell Biology. Body composition measurements were performed at the Vanderbilt Mouse Metabolic Phenotyping Center which is supported by the NIH grant 5U2CDK059637. Triglyceride and cholesterol measurements were performed by the Vanderbilt University Medical Center Lipid Core supported by the NIH DRTC grant DK020593. Research in the laboratory of DAJ was supported by NIH grants R01DK097392, R01DK129340, and R01DK115620.

## Additional information

### Funding

| Funder | Grant reference number | Author |
|---|---|---|
| National Institutes of Health | R01DK097392 | Prasanna K Dadi<br>Matthew T Dickerson<br>Soma Behera<br>David A Jacobson |
| National Institutes of Health | R01DK129340 | Arya Y Nakhe<br>Prasanna K Dadi<br>Matthew T Dickerson<br>David A Jacobson |
| National Institutes of Health | R01DK115620 | Arya Y Nakhe<br>Prasanna K Dadi<br>Matthew T Dickerson<br>Jordyn R Dobson<br>David A Jacobson |

The funders had no role in study design, data collection and interpretation, or the decision to submit the work for publication.

### Author contributions

Arya Y Nakhe, Conceptualization, Resources, Data curation, Formal analysis, Investigation, Methodology, Writing – original draft, Writing – review and editing; Prasanna K Dadi, Data curation, Formal analysis, Supervision, Investigation, Methodology; Jinsun Kim, Data curation, Formal analysis,

Methodology; Matthew T Dickerson, Data curation, Software, Supervision, Investigation, Methodology, Writing – review and editing; Soma Behera, Data curation, Formal analysis, Investigation, Methodology; Jordyn R Dobson, Data curation, Formal analysis, Investigation, Methodology, Writing – review and editing; Shristi Shrestha, Jean-Philippe Cartailler, Software, Formal analysis, Methodology; Leesa Sampson, Resources, Methodology; Mark A Magnuson, Conceptualization, Resources, Methodology; David A Jacobson, Conceptualization, Resources, Formal analysis, Funding acquisition, Investigation, Methodology, Writing – original draft, Project administration, Writing – review and editing

## Author ORCIDs
Arya Y Nakhe ⓘ http://orcid.org/0000-0003-2192-5865
Mark A Magnuson ⓘ http://orcid.org/0000-0002-8824-6499
David A Jacobson ⓘ https://orcid.org/0000-0003-1816-5375

## Ethics
Animals were handled in compliance with guidelines approved by the Vanderbilt University Animal Care and Use Committee protocols (#M2200007-00).

Reviewer #1 (Public Review): https://doi.org/10.7554/eLife.89967.3.sa1
Reviewer #2 (Public Review): https://doi.org/10.7554/eLife.89967.3.sa2
Reviewer #3 (Public Review): https://doi.org/10.7554/eLife.89967.3.sa3
Author response https://doi.org/10.7554/eLife.89967.3.sa4

---

# Additional files

## Supplementary files
- Supplementary file 1. All mouse primer sequences utilized for qRT-PCR.
- MDAR checklist

## Data availability
RNA-sequencing data has been deposited and is available in NCBI GEO with accession ID GSE239566.

The following dataset was generated:

| Author(s) | Year | Dataset title | Dataset URL | Database and Identifier |
|---|---|---|---|---|
| Nakhe AY, Dadi PK, Kim J, Shrestha S, Cartailler J, Sampson L, Magnuson MA, Jacobson DA | 2023 | The MODY-associated TALK-1 L114P mutation causes islet α-cell overactivity and β-cell inactivity resulting in transient neonatal diabetes and glucose dyshomeostasis in adults | https://www.ncbi.nlm.nih.gov/geo/query/acc.cgi?acc=GSE239566 | NCBI Gene Expression Omnibus, GSE239566 |

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
