## [Editor Report · eLife assessment]

This study characterizes how a point mutation in the TALK-1 potassium channel, encoded by the KCNK16 gene, causes MODY diabetes. The mutation, L114P, causes a gain-of-function to increase K+ currents and inhibit glucose-stimulated insulin secretion. Increased glucagon likely results from paracrine effects in the islets. The data are **convincing** and the work will be **valuable** for understanding islet function.

---

## [Referee Report · Reviewer #1 (Public Review)]

Summary:

This paper focuses on the effects of a L114P mutation in the TALK-1 channel on islet function and diabetes. This mutation is clinically relevant and a cause of MODY diabetes. This work employs a mouse model with heterozygous and homozygous mutants. The homozygous mice are homozygous lethal from severe hyperglycemia. The work shows that the mutation increases K+ currents and inhibits insulin secretion. This is a very nice paper with mechanistic insight and clear clinical importance. It is generally well written and the data is well presented.

Comments on revision:

I have no further comments to add at this time. The authors have adequately addressed my concerns.

---

## [Referee Report · Reviewer #2 (Public Review)]

Summary:

This work follows previous work from the group where they have demonstrated the role of TASK1 in the regulation of glucose stimulated insulin secretion. Moreover, a recent study links a mutation in KCNK16, the gene encoding TALK-1 channels to MODY. Here the authors have constructed a mouse model with the specific mutation (TALK-1 L114P mutation) and investigated the phenotype. They have to perform a couple of breeding tricks to find a model that is lethal in adult which might complicate the conclusions, however, the phenotype of the heterozygote model used have a MODY-like phenotype. The study is convincing and solid.

Strengths:

(1) The work is a natural follow-up from previous studies from the groups.

(2) The authors present convincing and solid data that in the long perspective will help patients with this mutations.

(3) Both in vivo and in vitro data are presented to give the full picture of the phenotype.

(4) Data from both female and male mice are presented.

Weaknesses:

The authors have answered all my comments in the revised version and I find no more weaknesses. Some questions still remain but have been clearly discussed in the new version of the manuscript.

---

## [Referee Report · Reviewer #3 (Public Review)]

Summary

The L114P gain of function mutation in the K2P channel TALK-1 encoded by KCNJ16 has been associated with maturity-onset diabetes of the young (MODY). In this study, Nakhe et al. generated mice carrying L114P TALK-1 and evaluated the impact of the mutation on pancreatic islet functions and glucose homeostasis. The authors report that the mutation increases neonatal lethality, owing to hyperglycemia caused by a lack of glucose-stimulated Ca2+ influx and insulin secretion. Adult mutant mice showed glucose intolerance and fasting hyperglycemia, which is attributed to blunted glucose-stimulated insulin secretion as well as increased glucagon secretion. Interestingly, male mice were more affected than female mice. Islets from adult mutant mice were found to have reduced Ca2+ entry upon glucose stimulation but also enhanced IP3-induced ER Ca2+ release, consistent with previous studies from the group showing a role of TALK-1 in ER Ca2+ homeostasis. Finally, comparison of bulk RNA sequencing results from WT and mutant islets revealed altered expression of genes involved in β-cell identify, function and signaling, which also contributes to the observed islet dysfunction.

Strengths

This is a well-executed and rigorous study that will be of great interest to the diabetes and islet biology communities. The findings provide convincing evidence supporting a causal role of the L114P gain of function TALK-1 mutation in glucose-stimulated insulin secretion defects and diabetes. The neonatal diabetes phenotype and the gender difference uncovered by the study have important clinical implications. The complexity of TALK-1 expression and hormone secretion in different endocrine cell types and how it impacts glucose homeostasis is elegantly illustrated in the L114P TALK-1 mouse model. The authors carefully and thoroughly addressed limitations of their study and discussed future directions. The importance of TALK-1 in β-cell and islet function demonstrated by this study will prompt future efforts targeting this important channel for diabetes treatment.

---

## [Author Response]

The following is the authors’ response to the original reviews.

**Reviewer #1:**
I have only a few comments that I think will improve the manuscript and help readers better appreciate the context of the reported results.

We would like to thank the Reviewer for their time in reviewing our manuscript. We appreciate the helpful feedback and assistance in ensuring the highest quality publication possible.

One paradox, that the authors point out, is that the drastic effects of TALK-1 L114P on plasma membrane potential do not result in a complete loss of insulin secretion. One important consideration is the role of intracellular stores in insulin secretion at physiological levels of hyperglycemia. This needs to be discussed more thoroughly, especially in the light of recent papers like Postic et al 2023 AJP and others. The authors do show an upregulation of IP3-induced Ca release. It is not clear whether they think this is a direct or indirect effect on the ER. Is there more IP3? More IP3R? Are the stores more full?

The reviewer brings up an important point. Although we see a significant reduction in glucose-stimulated depolarization in most islets from TALK-1 L114P mice, some glucosestimulated calcium influx is still present (especially from female islets); this suggests that a subset of islet β-cells are still capable of depolarization. Because our original membrane potential recordings were done in whole islets without identification of the cell type being recorded, we have now repeated these electrical recordings in confirmed β-cells (see Supplemental figure 6). The new data shows that 33% of TALK-1 L114P β-cells show action potential firing in 11 mM glucose, which would be predicted to stimulate insulin secretion from a third of all TALK-1 L114P β-cells; this could be responsible for the remaining glucosestimulated insulin secretion observed from TALK-1 L114P islets. However, ER calcium store release could also allow for some of the calcium response in the TALK-1 L114P islets. We have now detailed this in the discussion; this now details the Postic et. al. study showing that glucose-stimulated beta-cell calcium increases involve ER calcium release as it occurs in the presence of voltage-dependent calcium channel inhibition. Future studies can assess this using SERCA inhibitors and determining if glucose-stimulated calcium influx in TALK-1 L114P islets is lost. We also find that muscarinic stimulated calcium influx from ER stores is greater in TALK-1 L114P mice. We currently do not have data to support the mechanism for this enhancement of muscarinic-induced islet calcium responses from islets expressing TALK1 L114P. Our hypothesis is that greater TALK-1 current on the ER membrane is enhancing ER calcium release in response to IP3R activation. There is an equivalent IP3R expression in control and TALK-1 L114P islets based on transcriptome analysis, which is now included in the manuscript. However, whether there is greater IP3 production, greater ER calcium storage, and/or greater ER calcium release requires further analysis. Because this finding was not directly related to the metabolic characterization of this TALK-1 L114P MODY mutation, we are planning to examine the ER functions of TALK-1L114P thoroughly in a future manuscript.

The authors point to the possible roles of TALK-1 in alpha and delta cells. A limitation of the global knock-in approach is that the cell type specificity of the effects can't easily be determined. This should be more explicitly described as a limitation.

We thank the reviewer for this suggestion and have added this to the discussion. This is now included in a paragraph at the end of the discussion detailing the limitations of this manuscript.

The official gene name for TALK-1 is KCNK16. This reviewer wonders whether it wouldn't be better for this official name to be used throughout, instead of switching back and forth. The official name is used for Abcc8 for example.

We thank the reviewer for this suggestion and have revised the manuscript to include Kcnk16 L114P. The instances of TALK-1 L114P that remain in the manuscript are in cases where the text specifically discusses TALK-1 channel function.

There are several typos and mistakes in editing. For example, on page 5 it looks like "PMID:11263999" has not been inserted. I suggest an additional careful proofreading.

We have revised this reference, thoroughly proofread the revised manuscript, and corrected typos.

The difference in lethality between the strains is fascinating. Might be good to mention other examples of ion channel genes where strain alters the severe phenotypes? Additional speculation on the mechanism could be warranted. It also offers the opportunity to search for genetic modifiers. This could be discussed.

We thank the reviewer for this suggestion and have added details on mutations where strain alters lethality.

The sex differences are interesting. Of course, estrogen plays a role as mentioned at the bottom of page 16, but there have been more involved analyses of islet sex differences, including a recent paper from the Rideout group. Is there a sex difference in the islet expression of KCNK16 mRNA or protein, in mice or humans?

We thank the reviewer for the important comments on the TALK-1 L114P sex differences. We have revised the manuscript to include greater discussion about female β cell resilience to stress, which may allow greater insulin secretion in the presence of the TALK-1 L114P channels; this is based on the Brownrigg et. al. study pointed out by the reviewer (PMID: 36690328). Because these sex differences in islet function were examined in mice, we looked at KCNK16 expression in mouse beta-cells. While there is a trend for greater KCNK16 expression in sorted male beta-cells (average RPKM 6296.25 +/-953.84) compared to sorted female beta-cells (5148.25 +/- 1013.22). Similarly, there was a trend toward greater KCNK16 expression in male HFD treated mouse beta-cells (average RPKM 8020.75 +/- 1944.41) compared to female HFD treated mouse beta-cells (average RPKM 7551 +/- 2952.70). We have now added this to the text.

Page 15-16 "Indeed, it has been well established that insulin signaling is required for neonatal survival; for example, a similar neonatal lethality phenotype was observed in mice without insulin receptors (Insr-/-) where death results from hyperglycemia and diabetic ketoacidosis by P3 (40)." Formally, the authors are not examining insulin signaling. A better comparison is that of the Ins1/Ins2 double knockout model of complete hypoinsulinemia.

We thank the reviewer for suggesting this as the appropriate comparison model and have now revised the manuscript to detail the 48-hour average life expectancy of Ins1/Ins2 double knockout mice (PMID: 9144203).

There are probably too many abbreviations in the paper, making it harder to read by nonspecialists. I recommend writing out GOF, GSIS, WT, K2P, etc.

We thank the reviewer for this suggestion and have revised the manuscript to reduce the use of most abbreviations.

**Reviewer #2:**

We would like to thank the Reviewer for their time in reviewing our manuscript. We appreciate the helpful feedback and assistance in ensuring the highest quality publication possible. We have thoroughly addressed all the reviewer’s comments and revised the manuscript accordingly. These changes have strengthened the manuscript and are summarized below.

(1) The authors perform an RNA-sequencing showing that the cAMP amplifying pathway is upregulated. Is this also true in humans with this mutation? Other follow-up comments and questions from this observation:a) Will this mean that the treatment with incretins will improve glucose-stimulated insulin secretion and Ca2+ signalling and lower blood glucose? The authors should at least present data on glucose-stimulated insulin secretion and/or Ca2+ signalling in the presence of a compound increasing intracellular cAMP.b) Will an OGTT give different results than the IPGTT performed due to the fact that the cAMP pathway is upregulated?c) Is the increased glucagon area and glucagon secretion a compensatory mechanism that increases cAMP? What happens if glucagon receptors are blocked?

We thank the reviewer for the suggestions. Although cAMP pathways were upregulated in the TALK-1 L114P islets, the changes in expression were only modest as examined by qRTPCR. Thus, we are not sure if this plays a role in secretion. For humans with this mutation, there have been such a small number of patients and no islets isolated from these patients. Therefore, we are unaware if the cAMP amplifying pathway is upregulated in humans with the MODY associated TALK-1 L114P mutation. We have performed the suggested experiment assessing calcium from TALK-1 L114P islets in response to liraglutide (see Supplemental figure 10); there was no liraglutide response in TALK-1 L114P islets. We have also performed the OGTT experiments as suggested and these have now been added to the manuscript (see Supplemental figure 3). We do not believe that the increased glucagon is a compensatory response, because: 1. TALK-1 deficient islets have less glucagon secretion due to reduced SST secretion (see PMID: 29402588); 2. There is no change in insulin secretion at 7mM glucose, however, glucagon secretion is significantly elevated from islets isolated from TALK-1 L114P mice; 3. TALK-1 is highly expressed in delta-cells, and in these cells TALK-1 L114P would be predicted to cause significant hyperpolarization and significant reductions in calcium entry as well as SST secretion. Thus, reduced SST secretion may be responsible for the elevation of glucagon secretion. We plan to investigate delta-cells within islets from TALK-1 L114P mice in future studies to determine if changes in SST secretion are responsible for the elevated glucagon secretion from TALK-1 L114P islets.

(2) The performance of measurements in both male and female mice is praiseworthy. However, despite differences in the response, the authors do not investigate the potential reason for this. Are hormonal differences of importance?

We thank the reviewer for this important point. It is indeed becoming clear that there are many differences between male and female islet function and responses to stress. Thus, we have revised the manuscript to include greater discussion about these differences such as female β cell resilience to stress, which may allow greater insulin secretion in the presence of the TALK-1 L114P channels; this is based on the Brownrigg et. al. study pointed out by reviewer 1 (PMID: 36690328). While the differences in islet function and GTT between male and female L114P mice are clear, they both show diminished islet calcium handling, defective hormone secretion, and development of glucose intolerance. This manuscript was intended to demonstrate how the MODY TALK-1 L114P causing mutation caused glucose dyshomeostasis, which we have determined in both male and female mice. The mechanistic determination for the differences between male and female mice and islets with TALK-1 L114P could be due to multiple potential causes (as detailed in PMID: 36690328), thus, we believe that comprehensive studies are required to thoroughly determine how the TALK-1 L114P mutation differently impacts male and female mice and islets, which we plan to complete in a future manuscript.

(3) MINOR: Page 5 .." channels would be active at resting Vm PMID:11263999.." The actual reference has not been added using the reference system.

We thank the reviewer for noticing this mistake, which has now been corrected.

**Reviewer #3:**
The manuscript is overall clearly presented and the experimental data largely support the conclusions. However, there are a number of issues that need to be addressed to improve the clarity of the paper.

We would like to thank the Reviewer for their time in reviewing our manuscript. We appreciate the helpful feedback and assistance in ensuring the highest quality publication possible. We have thoroughly addressed all the reviewer’s comments and revised the manuscript accordingly. These changes have strengthened and improved the clarity of the manuscript.

Specific comments:(1) Title: The terms "transient neonatal diabetes" and "glucose dyshomeostasis in adults" are used to describe the TALK-1 L114P mutant mice. Transient neonatal diabetes gives the impression that diabetes is resolved during the neonatal period. The authors should clarify the criteria used for transient neonatal diabetes, and the difference between glucose dyshomeostasis and MODY. Longitudinal plasma glucose and insulin data would be very informative and help readers to follow the authors' narrative.

We appreciate the helpful comment and have added longitudinal plasma glucose from neonatal mice to address this (see Supplemental figure 2). The new data now shows the TALK-1 L114P mutant mice undergo transient hyperglycemia that resolves by p10 and then occurs again at week 15. Insulin secretion from P4 islets is also included that shows that male animals homozygous for the TALK-1 L114P mutation have the largest impairment in glucosestimulated insulin secretion, followed by male heterozygous TALK-1 L114P P4 islets that also have impaired insulin secretion (see Figure 1). The amount of hyperglycemia correlates with the defects in neonatal islet insulin secretion.

(2) Another concern for the title is the term "α-cell overactivity." This could be taken to mean that individual α-cells are more active and/or that there are more α-cells to secrete glucagon. The study does not provide direct evidence that individual α-cells are more active. This should be clarified.

We appreciate the helpful comment and have revised the manuscript title accordingly.

(3) In the Introduction, it is stated that because TALK-1 activity is voltage-dependent, the GOF mutation is less likely to cause neonatal diabetes, yet the study shows the L114P TALK-1 mutation actually causes neonatal diabetes by completely abolishing glucose-stimulated Ca2+ entry. This seems to imply TALK-1 activity (either in the plasma membrane or ER membrane) has more impact on Vm or cytosolic Ca2+ in neonates than initially predicted. Some discussion on this point is warranted.

These are important points and we have added details to the discussion about this. For example, the discussion now states that, “This suggests a greater impact of TALK-1 L114P in neonatal islets compared to adult islets. Future studies during β-cell maturation are required to determine if TALK-1 activity is greater on the plasma membrane and/or ER membrane compared with adult β-cells.” The introduction has also been revised to clarify the voltagedependence of TALK-1.

(4) What is the relative contribution of defects in plasma membrane depolarization versus ER Ca2+ handling on defective insulin secretion response?

We thank the reviewer for bringing up this important point. TALK-1 L114P islets show blunted glucose-stimulated depolarization and glucose-stimulated calcium entry, however, the L114P islets show equivalent Ca2+ entry as control islets in response high KCl (Figure 5GH). As the KCl stimulated Ca2+ influx is similar between control and TALK-1 L11P islets, this indicates that plasma membrane TALK-1 L114P has a hyperpolarizing role that significantly blunts glucose-stimulated depolarization and reduces activation of voltage-dependent calcium channels. We have further tested this by looking at glucose-stimulated β-cell membrane potential depolarization in TALK-1 L11P islets, which is significantly blunted (Figure4 A and B; Supplemental figure 6). However, 33% of TALK-1 L11P β-cells showed glucose-stimulated electrical excitability (Supplemental figure 6), which likely accounts for the modest GSIS from TALK-1 L11P islets. New data has also been included showing that KCl stimulation causes a significant depolarization of β-cells from TALK-1 L11P islets (Supplemental figure 6). Because plasma membrane TALK-1 L114P is largely responsible for the hyperpolarized membrane potential and blunted glucose-stimulated Ca2+ entry, this suggests that TALK-1 L11P on the plasma membrane is primarily responsible for the altered insulin secretion. The discussion has been revised to reflect this.

(5) The Jacobson group has previously shown that another K2P channel TASK-1 is also involved in ER Ca2+ homeostasis and that TASK inhibitors restored ER Ca2+ in TASK-1 expressing cells. Is TASK-1 expressed in β-cell ER membrane? Can the mishandling of Ca2+ caused by TALK-1 L114P be reversed by TASK-1 inhibitors?

We thank the reviewer for bringing up this important point in relation to ER calcium handling by K2P channels. We have found that TASK-1 channels expressed in alpha-cells enhance ER calcium release and that inhibitors or TASK-1 channels elevate alpha-cell ER calcium storage. We did not observe any significant changes in the gene (Kcnk3) encoding TASK-1 between islets from control or TALK-1 L11P mice, which has now been added to the manuscript. However, because the TALK-1 L11P-mediated reduction of glucose-stimulated depolarization and inhibition of calcium entry are both prevented in the presence of high KCl (see Figure X); this strongly suggests that TALK-1 L114P K+ flux at the membrane is hyperpolarizing the membrane potential and limiting depolarization and calcium entry. This suggests that TALK-1 L114P control of ER calcium handling is not the primary contributor to the blunted glucose-stimulate calcium handling. Furthermore, acetylcholine stimulation of islets from both control and TALK-1 L114P islets elicited ER calcium release, which indicates that for the most part ER calcium release is still responsive to cues that control release, but they are altered. Taken together this suggests that the TALK-1 L114P impact on ER calcium is not the primary mediator of blunted glucose-stimulated islet calcium entry and insulin secretion.

(6) The electrical recording experiments were conducted using whole islets. The authors should comment on how the cells were identified as β-cells, especially in mutant islets in which there is an increased number of α-cells.

The reviewer brings up an important point. As indicated, the original membrane potential recordings were conducted using whole islets. While the recorded cells could mostly be βcells based on mouse islets typically containing >80% β-cells, there is a possibility that some of the cells included in these recordings were α-cells or δ-cells (especially because of the noted α-cell hyperplasia in TALK-1 L114P islets). Thus, we have now included data from bcells that were identified with an adenoviral construct containing a rat insulin promoter driving a fluorescent reporter. This allowed the fluorescent β-cells to be monitored with electrophysiological membrane potential recordings. The new data (see Supplemental figure 6) shows a significant reduction in glucose-stimulated depolarization in 67% of β-cells with the L114P mutation compared to controls.

Minor:(1) Some references need formatting.

The references have been revised accordingly.

(2) Please define glucose-stimulated phase 0 Ca2+ response for non-expert readers.

This has been defined accordingly.

(3) Page 14 bottom: The sentence "Unlike the only other MODY-associated.........., TALK-1 is not inhibited by sulfonylureas" seems out of place and lacks context.

We thank the reviewer for this suggestion and have deleted this sentence.

(4) Figure 6: It would be helpful to provide a protein name for the genes shown in panel D.

The protein names for the genes have now been included in the discussion of these genes.